



# 1 40Ar/39Ar age constraints on the formation of fluid-rich quartz veins
# 2 from the NW Rhenohercynian zone (Rursee area, Germany)

Akbar Aydin Oglu Huseynov[1*], Jan R. Wijbrans[1], Klaudia F. Kuiper[1] & Jeroen van der Lubbe [1]
[1]Vrije Universiteit Amsterdam, De Boelelaan 1085, 1081HV Amsterdam, the Netherlands
*Correspondence to*: Akbar Aydin Oglu Huseynov (a.huseynov@vu.nl)
**Abstract.**  The late Palaeozoic Variscan orogeny (~350 Ma) dictates a significant part of the subsurface geology in north-
western and central Europe. Our focus is particularly on veining that occurred in metamorphosed sedimentary rocks that are
affected by this orogeny. Vein minerals serve as repositories for documenting the origin of subsurface fluid flows and
dynamics, and dating them provides crucial insight into the timing of orogenic and possible reactivation events. The Rursee
area (Rhenish Massif, Germany) that is part of the Variscan foreland zone on the Avalonia micro-continent represents a key
locality for studying Variscan quartz vein formation.  Based on structural grounds, the two different groups/types of Rursee
quartz veins have been linked with the early stages of Variscan, but their absolute ages are still unknown.
The aim of this study is to date these quartz veins using the 40Ar/39Ar stepwise crushing method based on the radioactive decay
of 40K dissolved in high salinity fluid inclusions (FIs). We obtained Jurassic to Cretaceous ages, and the isotopic analysis of
argon gases revealed that the fluid-rich quartz fractions release 39Ar in two distinct phases. Regardless of quartz veins FIs
salinity, stepwise crushing provides apparent K/Cl >1. Electron Probe Micro Analyser data confirm the presence of the K
(39Ar) in the K-bearing mineral inclusions (e.g., sericite, mica, and chlorite) and in microcracks and possibly in the crystal
lattice of quartz.
K-bearing mineral inclusions and/or crystal lattice of quartz, which form in the Variscan-origin vein fractures, provide a
plausible explanation for the young apparent isotopic ages. The presence of the quartz sub-grains may suggest that obtained
ages are likely to reflect post-Variscan reactivation-recrystallisation due to tectonic activity or its cooling moment during the
Jurassic-Cretaceous period rather than the original Variscan vein formation.
This study emphasizes the complexities of isotopic dating of FIs, as well as the importance of careful interpretation of such
data, especially in cases where different K-bearing mineral inclusions and/or radiogenic argon from crystal lattice obscure the
initial FIs signal.

## 26  1 Introduction

Quartz veins are abundant in metamorphosed terranes and sedimentary basins filled with siliciclastic sediments, witnessing
significant fluid movement during diagenesis and metamorphism (Yardley, 1983; Mullis et al., 1994; Cartwright & Buick,
2000; Oliver & Bons, 2001). Increase in both temperature and pressure during burial diagenesis, orogenesis and deformation





cause sedimentary and volcanic rocks to lose their volatile components and to release warm fluids, which cumulate minerals
in fractures and faults (Baumgartner & Ferry, 1991; Yardley & Bottrell, 1993; Oliver & Bons, 2001; Cox, 2007). These often
saline fluids contain, among others, $KCl_{(aq)}$ or $K_2CO_{3 (aq)}$ (Rauchenstein-Martinek et al., 2014), which are partly precipitated
during crystallisation of minerals in veins or as inclusions in these minerals (Sterner et al., 1988). One of the isotopes of
potassium, $^{40}K$, is radioactive and can be used for K-Ar or its derivative $^{40}Ar/^{39}Ar$ dating. Progressive crushing techniques
enable to liberate gasses from fluid inclusions (FIs), mineral inclusions and/or crystal lattice for the age determination of
geological events provided that K-concentrations are high enough (Qiu & Dai, 1989; Turner & Bannon, 1992; Turner & Wang,
1992; Qiu, 1996; Kendrick et al., 2001; Qiu & Wijbrans, 2006; Kendrick et al., 2006; Qiu & Wijbrans, 2008; Qiu & Jiang,
2007; Jiang et al., 2012; Bai et al., 2013; Liu et al., 2015). This method does not only define an age, but also quantifies the
ratio of noble gases (e.g. $^{39}Ar_K /^{37}Ar_{Ca,}$ $^{39}Ar_K /^{38}Ar_{Cl}$) derived from Ca, K and Cl, respectively, that have been formed during
neutron radiation prior to analysis. The $^{39}Ar_K/^{38}Ar_{Cl}$ provides important information on the composition of parental fluids and
their sources (Sumino et al., 2011; Cartwright et al., 2013). In addition to fluid composition and provenance studies (Kelley et
al.,1986; Turner & Bannon, 1992: Kendrick et al., 2001, 2006), the initial $^{40}Ar/^{36}Ar$ values of FIs in quartz can considerably
vary and may be used to differentiate between meteoric-sourced water (~298.6) (Ballentine et al., 2002; Ozima & Podosek,
2002) and deeper crustal or mantle-derived fluids (>10,000; MORB >40,000) (Burnard et al., 1997).
In order to accurately determine the age of FIs in quartz veins using the $^{40}Ar/^{39}Ar$ stepwise crushing method or the source of
the fluid based on $^{40}Ar/^{36}Ar$ ratios, it is necessary to consider three distinct components of $^{40}Ar$, namely (1) radiogenic $^{40}Ar_R$
or $^{40}Ar^*$, which is produced in the sample itself through the radioactive decay of $^{40}K$, and (2) $^{40}Ar$ that was initially trapped in
the fluid inclusion, either as (2a) atmospheric $^{40}Ar_A$ or (2b) excess $^{40}Ar_E$. The presence of $^{40}Ar_E$ in FIs creates a significant
challenge to determining accurate vein formation ages using the $^{40}Ar/^{39}Ar$ technique (Rama et al., 1965), although isochron
diagrams might help to overcome this issue (McKee et al., 1993; Qiu, 1996; Qiu et al., 2002). In addition to $^{40}Ar_E$, the origin
of $^{39}Ar_K$ (or K content) has been a topic of debate, with the possibility that the $^{39}Ar_K$ (and thus K) may come from the dissolved
salts in FIs, leaking from the crystals lattice during crushing (Kendrick et al., 2011), and/or from any K-bearing mineral
inclusions trapped inside the crystals (Qiu & Wijbrans, 2006; Kendrick, 2007; Qiu & Wijbrans, 2009; Kendrick & Phillips,

54  2009).

This study aims i) to determine the absolute age of quartz vein formation by analysing FIs using the stepwise crushing $^{40}Ar/^{39}Ar$
dating method, ii)  to elucidate the location of K in the vein minerals (e.g., FIs, mineral inclusions, and/or crystal lattice) and
iii) to identify when different K sources release their $^{39}Ar_K$ through the examination of released argon gases during the crushing
process and geochemical analysis of quartz mineral samples using an Electron Probe Micro Analyser (EPMA).
Quartz samples were obtained from an outcrop near the Rursee in the upper reaches of the Rur river in the North Eifel region
of Western Germany. Detailed structural investigations of this area have been previously conducted by Van Noten et al. (2007),
who differentiated quartz veins into two groups. The older generation of quartz veins, the so-called bedding normal veins
(BNVs) is assigned to the early stages of the Variscan orogeny, whereas the second group, comprising bedding parallel veins
(BPVs), is linked to the main stage of the Variscan orogeny. Absolute $^{40}Ar/^{39}Ar$ ages of FIs representing the age of quartz vein



formation would allow us to better constrain the structural evolution and subsurface fluid flow during the Variscan orogeny in
north-western Europe. Reliable $^{40}Ar/^{39}Ar$ age constraints of quartz vein formation would provide the opportunity to understand
the timing and evolution of mountain building in analogue fold-and-thrust belts.

## 1.1 Geological Setting

The Rhenohercynian fold-and-thrust belt, part of the Variscan, is primarily located in the Rhenish Massif in Germany and
extends westward into the Ardennes, southwest England, and eastward to the Harz Mountains  (Kołtonik et al., 2018). The
Ardennes Allochthone (Fig. 1a), western part of Rhenish Massif, structurally comprises three main components: the Dinant
fold-and-thrust belt, the Lower Palaeozoic Inliers, and the High-Ardennes Slate Belt (HASB). The HASB primarily consists
of Lower Devonian metasediments including the Rurberg (upper Pragian) and Heimbach (upper Pragian to lower Emsian)
units.
For this study, quartz veins samples were collected near the Schwammenauel dam in the Rursee area of the North Eifel region,
Germany (Fig. 1b). The Rurberg and Heimbach units feature alternating layers of siltstones and fine- to coarse-grained
sandstones (Goemaere & Dejonghe, 2005), deposited in shallow marine to deltaic environments in the northern
Rhenohercynian Ocean (Oncken et al., 1999). The Early Devonian strata have accumulated to a total thickness of up to 7 km
due to rapid subsidence and deposition (Winterfeld, 1994) forming the Eifel syncline (Fig. 1b). These strata are overlain by a
~3 km thick sequence of Lower Lochkovian to Pragian deposits.
The late Carboniferous deformation of the Variscan foreland led to initial burial metamorphism (Mansy et al., 1999), with
prehnite-pumpelleyite facies similar to the anchizone conditions in the North Eifel area (Fielitz 1995), where temperatures
reached up to 220 °C (Littke et al., 2012). There is also evidence of the upward migration of warm fluids into the northern
Variscan front in Ardennes, driven by Variscan thrusting (Muchez et al., 2000; Schroyen & Muchez, 2000; Lünenschloss et
al., 2008).
Following the Variscan period, the Rhenish Massif has been affected by transpressional and transtensional deformation that
resulted in the formation of complex fault networks that host vein mineralization (Franzke & Anderle, 1995; Ziegler & Dèzes,
2005). During the Jurassic-Cretaceous period, the southern Rhenish Massif was periodically affected by hydrothermal
activities (Kirnbauer et al., 2012), as indicated by geochronological data for post-Variscan vein mineralization (Bonhomme et
al., 1983; Mertz et al., 1986; Bähr, 1987; Jakobus, 1992; Hein & Behr, 1994; Klügel, 1997; Schneider & Haack, 1997;
Glasmacher et al., 1998; Schneider et al., 1999; Chatziliadou & Kramm, 2009).





**Figure 1: (a) Geological map with the Variscan frontal zone in the Ardenne-Eiffel region (study area marked with a red star). (b)**
**Geological map of the North Eiffel region (modified after Ribbert, 1992; Van Noten et al., 2011). The Lower Devonian layers overlay**
**metamorphic deposits of the Lower Palaeozoic Stavelot-Venn Inlier. These layers have been locally distorted in the Monschau Shear**
**Zone (MSZ), as documented by Fielitz (1992). Triassic sediments overlay the Lower Devonian layers in the eastern region. The**
**sample location, indicated by a green star, is situated next to the Rursee reservoir, which is near the Schwammenauel dam. Below,**
**cross-section illustrates the continuous northwest-southeast trending overturned folds that are characteristic of the North Eifel zone.**





The $^{40}$Ar/$^{39}$Ar study targets the BNVs and BPVs (Fig. 2), which formed in low-grade metamorphosed (prehnite-pumpelleyite
facies) conditions as a result of the precipitates from warm fluids in fractures (Van Noten et al., 2008). The structural cross-
cutting relationships between these quartz vein generations suggest that they originated during different geological events (Van
Noten et al., 2008) revealing that BPVs are younger than BNVs. BNVs are found mostly within the competent psammite and
hardly occur in incompetent pelitic layers. This positioning suggests that BNVs formed during the early stages of the Variscan
orogeny, associated with the final burial phases of the Ardennes-Eifel basin (Sintubin et al., 2000; Urai et al., 2001; Van Noten
et al., 2008, 2009).
In contrast, BPVs follow the strata between the psammatic and pelitic layers as a result of the bedding-parallel slip caused by
flexural folding during the Variscan orogeny (Van Noten et al., 2008).
**Figure 2: Images of studied outcrop from the Rursee area. The image (a) presents the bedding normal veins (red lines), while (b)**
**shows the bedding parallel veins (red lines). Yellow lines indicate the bedding in both images.**





## 2 Material and methods

### 2.1 Quartz and inclusions in quartz minerals

A total of seven samples of different veins (3 BNVs and 4 BPVs) were collected from the Rursee outcrop for $^{40}Ar/^{39}Ar$ analysis (Table 1). Both vein types mainly consist of elongated-fibrous milky quartz grains that are characterised by syntaxial growth, whereby the growth starts from the wall of the veins towards the central part of the veins (Ramsay, 1986). The pelitic host rocks consist of sericite, illite, mica and chlorite. Chlorite is also abundant within the vein fractures and between the host rock and the vein wall.

Both quartz vein generations lack of primary FIs in the crystal growth zones and contain pseudo-secondary and secondary fluid inclusion assemblages (FIAs) (<10 µm) (Van Noten et al., 2011) in the sealed microcracks being perpendicular to crystal elongation (Fig 3.). The Rursee quartz vein samples yield average FIs homogenization temperatures (minimum trapping temperature, $T_h$) of ~135 ±25 °C and ~160 ±20 °C for BPV and BNV, respectively, with salinities of 3.5-8 eq. wt.% NaCl  In general, $T_h$ of pseudo-secondary and secondary FIs span an equally broad range of 110-180 °C (Van Noten et al., 2011).

### 2.2 Mineral separation

Prior to $^{40}Ar/^{39}Ar$ analysis, mineral separation was conducted at Vrije Universiteit Amsterdam (VU; The Netherlands). The bulk vein samples were crushed, washed, and cleaned in an ultrasonic bath for at least one hour to remove the adhering host rock contaminants from quartz grains. The samples were sieved into 250 and 500 µm fraction and dried in an oven at 60 °C. The samples were further separated by a custom-made system using an overflow centrifuge with conventional heavy liquids based on IJlst (1973) and Frantz magnetic separation (Porat, 2006). We used heavy liquids with a density of 2.62 g/cm³ and 2.64 g/cm³ to obtain fluid inclusion-rich fraction of quartz grains (ρ= 2.62-2.64 g/cm³). The fraction was rinsed with acetone, dried, and further sieved to separate the 400-500 µm grain size range. From this fraction, only the purest quartz grains were hand-picked under binocular microscope for $^{40}Ar/^{39}Ar$ dating.

**Figure 3: Fluid inclusions in quartz veins under optical microscopy. (a) Image of BNVs under cross-polarizer light microscopy. Both (a.1.1) and (a.2.1) are the zoom of (a.1) and (a.2) images, respectively, indicating pseudo-secondary fluid inclusion assemblages (FIAs) (white arrow). (b) Cross polarizer images of the BPV sample under microscopy. (b.1.1) and (b.2.1) images are secondary and pseudo-secondary fluid inclusion (respectively) -focused areas, which are zooms of the (b.1) and (b.2) images, respectively. The white arrows represent the FIAs. Both generations of quartz veins have FIAs that are present in sealed microcracks rather than in crystal growth zones.**





### 2.3 $^{40}$Ar/$^{39}$Ar stepwise crushing

Fluid-rich quartz grains (400-500 μm; 2.62-2.64 g/cm$^3$) were carefully selected under a binocular zoom microscope, and a quantity of 200-270 mg of material was packed in aluminium foil and placed in 20 mm ID - 22mm OD aluminium cups. Drachenfels (DRA-2) sanidine standard was loaded between each set of three samples to monitor the neutron flux. The samples were irradiated at Oregon State University (USA) using the CLICIT (Cadmium-Lined In-Core Irradiation Tube) facility for 12 hours (batch VU123). After irradiation, standards were placed in 2 mm copper planchet holes for single grain fusion analysis and vacuum pre-baked at 250 °C. The samples were then placed in an ultra-high vacuum system, baked at 120 °C, and connected with hot NP10 and ST172 getters, Ti getter sponge at 400 °C, and a cold trap at -70 °C. The standards were fused with a Synrad 48–5 $CO_2$ continuous-wave laser fusion system.

The samples were crushed in an in-house developed and built crusher consisting of a stainless-steel tube (height: 18 cm, outer diameter: 1.8 cm) that has a spherical curve on its interior base and a magnetic stainless-steel pestle (height: 5 cm, diameter: 1.6 cm, weight: ~69.5 g) with rounded tips with a slightly narrower outer radius. These geometries allow optimisation of the impact on the sample while crushing. Once a split of the sample (~30 mg of quartz grains) was loaded into the crusher tube, the pestle was carefully relocated to the bottom of the tube to avoid crushing the sample. The crush tube, the pestle, and the sample were baked overnight at 250 °C. The pestle was dropped into a free-fall state using an external electromagnet with a frequency of 1 Hz controlled by an adjustable power supply and pulse generator to crush the sample. The pestle was dropped from a height of ~3, ~4 or ~5 cm *in vacuo*. Subsequently, the gases emitted from FIs in the fragmented quartz sample were analysed. To obtain a sufficient amount of argon in the mass spectrometer, the number of pestle drops per extraction step and drop height were systematically increased during the experiment, with a maximum of 999 drops per analysis (in total, ~40000 cumulated pestle drops per experiment).

The gas released from the samples and standards was analysed isotopically using a ThermoFisher Scientific Helix MC+ mass spectrometer. The Helix MC+ mass spectrometer is a 5 collector channel instrument, equipped with a total of 10 collectors, a Faraday collector optionally fitted with a $10^{12}$ Ohm or $10^{13}$ Ohm resistor amplifier and a compact discrete dynode secondary electron multiplier (CDD-SEM) collector on each collector channel. Five collectors can be used at the same time to simultaneously collect the beam intensity signals of the 5 isotopes of argon. The H2-Faraday collector is employed for the detection of $^{40}$Ar using a $10^{13}$ Ohm amplifier. Similarly, the H1- CDD collector is used for the measurement of $^{39}$Ar (H1 Faraday was used for the runs on DRA-2 sanidine standard because of the higher $^{39}$Ar signal), the AX-CDD collector for $^{38}$Ar, the L1-CDD collector for $^{37}$Ar, and the L2-CDD collector for $^{36}$Ar.

Line blanks were measured after every three to four unknowns and subtracted from the succeeding sample data. A Gain calibration is done by correcting for gain relative to the beam intensity measured on the AX-CDD, using measurements of ~50 fA ($^{40}$Ar measured beam intensities) pipettes of air on each cup, and mass discrimination corrections are made by measuring a series of ~400 fA ($^{40}$Ar measured beam intensities) air pipettes roughly every 12 hours. Raw data were processed using the ArArCalc software (Koppers, 2002). Ages are calculated relative to Drachenfels (DRA-2) sanidine of 25.552 ± 0.078 Ma





(Wijbrans et al., 1995) which was recalibrated against Fish Canyon Tuff sanidine of 28.201 ± 0.023 Ma (Kuiper et al., 2008).
The decay constants of Min et al. (2000) are used. The atmospheric $^{40}Ar/^{36}Ar$ ratio of 298.56 ± 0.31 is based on Lee et al.
(2006). The correction factors for neutron interference reactions are (2.64 ± 0.02) x10$^{-4}$ for $(^{36}Ar/^{37}Ar)_{Ca}$, (6.73 ± 0.04) x10$^{-4}$
for $(^{39}Ar/^{37}Ar)_{Ca}$, (1.21 ± 0.003) x10$^{-2}$ for $(^{38}Ar/^{39}Ar)_K$, and (8.6 ± 0.7) x10$^{-4}$ for $(^{40}Ar/^{39}Ar)_K$. Gain correction factors and their
standard errors (±1SE) are 1.00162 ± 0.00028 for H2-Far, 0.97963 ± 0.00021 for H1-CDD,  0.99921 ± 0.00027 for L1-CDD
and 0.96163 ± 0.00064 for L2-CDD for data measured in 2022 (R2.1) and 1.00465 ± 0.00031 for H2-Far, 0.97033 ± 0.00027
for H1-CDD,  0.99824 ± 0.00033 for L1-CDD, and 0.96309 ± 0.00070 for L2-CDD for data measured in 2023 (R1-R6). The
K/Cl ratios are calculated by K/Cl = β × $^{39}Ar/^{38}Ar$ with β = 0.06 derived from K/Cl = ~18.7 in GA1550 and $^{39}Ar_K/^{38}Ar_{Cl}$ =
~316 for a 12-hour irradiation at the OSU Triga CLICIT facility. All errors are quoted at the 2σ level and include all analytical
uncertainties (Table 1).
Note that it is not possible to directly correct the crushing blank because we cannot perform the exact experiment without
crushing sample material. We tested the blanks for each tube without sample material, following the identical procedures used
for real experiments. With this approach, we have direct metal-to-metal contact during pestle drops, which might not be fully
representative of a real sample. We did observe a substantial increase in background, with a higher number of drops and a
higher drop level. Importantly, the composition of this blank is similar to that of atmospheric argon. Therefore, we follow the
approach that the $^{40}Ar$ signal derived from the line blank (measured every 3-4 unknows where we mimic the sample
experiment, but without the crushing / pestle drops) is subtracted from the measured $^{40}Ar$ intensity. The real blank has an
atmospheric $^{40}Ar/^{36}Ar$ ratio and is incorporated in the air corrections, leading to a lower radiogenic $^{40}Ar^*$ if the real blanks are
relatively high.
**2.4 Electron Probe Microanalysis (EPMA)**
Quartz grains of sub-samples that were analysed for $^{40}Ar/^{39}Ar$ were mounted in epoxy resin and carbon coated for the JEOL
JXA-8530F hyperprobe field emission electron probe microanalyzer (EPMA) at Utrecht Universiteit (UU; The Netherlands)
to define the elemental compositions of 1) the host quartz, 2) minerals that are present in FIs, filled cavities, or fractures, and
3) mineral inclusions in the quartz. For this analysis, an accelerating voltage of 15 kV and a beam current of 8 nA for host rock
(quartz) and 7 nA for mineral inclusions are used with beam sizes of 10 μm and 1 μm, respectively. The elements analysed are
Si, Ti, Al, Fe, Mn, Ca, Na, K, P, Cl, F, Ba, and Zr.  The data are calibrated using Icelandic rhyolite glass (ATHO-G) and basalt
glass (KL2-G) standards that were both measured with a beam size of 10 μm, and multiple times before and after measurements
of the samples.





## 3 Results

The age spectra of the *in vacuo* stepwise crushing of the quartz samples are plotted in Figure 4. All samples show typical release patterns with unrealistically old ages (>6 Ga) in the initial 10 % of $^{39}Ar_K$ released. Note that samples Rursee 1a BNV and Rursee 1b BNV are measured in two different experiments on subsets from the same irradiated sample, yielding different results. For sample Rursee 1a BNV, a lighter pestle (68 g) has been used than for sample Rursee 1b BNV (69.5 g) and for all other samples.

The apparent ages of the spectra in samples Rursee 1b BNV, Rursee 2 BPV, Rursee 2.1 BNV, and Rursee 4 BPV exhibit a gradual decrease in age over the next 10 - 40 % of $^{39}Ar_K$ released, eventually stabilising at a more or less consistent age from ~80 to ~100 % $^{39}Ar_K$. Rursee 3 BPV, Rursee 5 BNV, and Rursee 6 BPV show comparable behaviour with, after the initial old ages, a decrease in age to a "pseudo-plateau" from ~15 % to ~40 % $^{39}Ar_K$ released, followed by a gradual decrease in age and a more or less uniform age in the >80 % released $^{39}Ar_K$ part of the spectrum. For these pseudo-plateaus, we arrive at averaged ages of ~84 Ma for Rursee 1b BNV, ~97 Ma for Rursee 2 BPV, ~117 Ma for Rursee 4 BPV, ~216 Ma for Rursee 2.1 BNV, ~190-200 Ma for Rursee 5 BNV, and Rursee 6 BPV, and ~560 Ma for Rursee 3 BPV. The ages of Rursee 2.1 BNV and Rursee 4 BPV correspond to the inverse isochron ages; however, due to significant uncertainty, the ages of other samples obtained from the average plateau age (Table 1).

The inverse isochrons (Fig. 5) confirm that the first part of all experiments is heavily affected by excess argon ($^{36}Ar/^{40}Ar$ ratios are much lower than atmospheric composition), followed by an increase in $^{36}Ar/^{40}Ar$ and $^{39}Ar/^{40}Ar$ ratios and clustering of data points on the reference line. The ages that we derive are based on the data points that cluster along the reference line in the isochrons in the final part of the age spectra. There is no systematic age difference between BNV and BPV.

All quartz samples release argon during *in vacuo* stepwise crushing with different isotopes of argon contributing to the gas release at different stages of the experiment. Figure 6 shows, for each step, the percentage (relative to total amount) of a specific isotope that is released through the experiment. All quartz samples are characterised by a release of most of the $^{36}Ar_{air}$ in the first 20 steps. $^{40}Ar^*$ and $^{38}Ar_{Cl}$ follow the pattern of $^{36}Ar_{air}$. The $^{39}Ar_K$ generally starts to increase after the first 20 analysing steps (~790 pestle drops from 3 cm height). At steps 30-35, we observe fluctuations in the data. These shifts are artefacts caused by increasing the drop height (from 3 to 4 cm at ~step 30 and from 4 to 5 cm at ~step 35) and adjusting the number of pestle drops. To prevent high signals, we started with a relatively low number of pestle drops at a higher drop height, yielding low signals, as observed as two troughs at ~step 30 and ~step 35 in all experiments. All quartz samples are low in $^{36}Ar_{air}$, $^{38}Ar_{Cl}$, and $^{40}Ar^*$ at the end of analysis compared to their total release. For $^{40}Ar^*$, we still measure a small, reliable signal, but this is obscured in Figure 6 due to the high signals in the first steps since we plot percentages of the total released $^{40}Ar$ per experiment. Note that huge amounts of excess $^{40}Ar$ (which is part of the $^{40}Ar^*$ signal) are released in the initial steps of the experiment and dominate the total percentage.





**Figure 4: The apparent plateau age of all quartz vein experiments. The red boxes focus on the last part of the age spectra, where apparent ages are more or less stable.**





Figure 5: Inverse isochrons of all quartz veins samples. Dark line corresponds to the atmospheric $^{36}Ar/^{40}Ar$, while pink line shows mean weighted.



**Figure 6: Released argon isotopes per analysing step relative to its total release. Note that the data are expressed against analysing step instead of the crushing step, and that the upper x-axis scaling (cumulative pestle drops) are neither linear, nor logarithmic (non-continuous scaling).**





**Table 1 Summary of $^{40}$Ar/$^{39}$Ar age spectra, including invers isochron data of all analysed quartz samples.**

| Locality | Rursee, outcrop near Schwammenauel dam (Germany) | | | | | | | |
|---|---|---|---|---|---|---|---|---|
| Rock type | Quartz veins | | | | | | | |
| Mineral | Quartz | | | | | | | |
| Sample ID | Rursee 1a BNV | Rursee 1b BNV | Rursee 2 BPV | Rursee 2.1 BNV | Rursee 3 BPV | Rursee 4 BPV | Rursee 5 BNV | Rursee 6 BPV |
| Sample ID Ar | R01a | R01b | R02 | R021 | R03 | R04 | R05 | R06 |
| GPS coordinate | Lat.: 50.63378406 Long.: 6.44191402 | | Lat.: 50.63377933 Long.: 6.44190753 | Lat.: 50.63388498 Long.: 6.44184657 | Lat.: 50.63418108 Long.:6.44176707 | Lat.: 50.6344143 Long.: 6.4418217 | Lat.: 50.63367794 Long.: 6.44201891 | Lat.: 50.63392217 Long.: 6.44181953 |
| Grain Size (µm) | 400 – 500 | | | | | | | |
| Density (g.cm$^{-3}$) | 2.62 - 2.64 | | | | | | | |
| Age (Ma) | 2843.9 | 83.9 | 96.6 | 144.9 | 559.9 | 129.3 | 192.6 | 200.9 |
| ±2σ analytical error + J error | ± 87.7 | ± 1.0 | ± 2.1 | ± 6.9 | ± 45.3 | ± 5.0 | ± 14.3 | ± 6.2 |
| ±2σ full external error | ± 95.9 | ± 2.0 | ± 2.9 | ±7.5 | ± 46.5 | ± 5.6 | ± 14.8 | ± 7.4 |
| MSWD | 42.8 | 1.6 | 3.34 | 4.34 | 6.05 | 5.39 | 2.01 | 0.37 |
| K/Ca | 0.32 | 1.54 | 14.28 | 5.33 | 0.285 | 3.21 | 0.75 | 3.48 |
| $^{40}$Ar/$^{36}$Ar inverse isochrone intercept | 3874 | 326 | 858 | 258 | 329 | 311 | 289 | 243 |
| ±2σ analytical error + J error | ± 7284.5 | ± 51.1 | ± 860.2 | ± 38.4 | ± 183.4 | ± 6.4 | ± 29.0 | ± 180.8 |
| Inverse isochrone age | - | 74.1 | 80.7 | 215.5 | 399.2 | 116.7 | 258.6 | 26.2 |
| ±2σ analytical error + J error | ± 5769.6 | ± 16.0 | ± 224.8 | ± 50.6 | ± 425.1 | ± 7.1 | ± 129.0 | ± 176.5 |
| ±2σ full external error | ± 5769.7 | ± 16.0 | ± 224.8 | ± 50.8 | ± 425.2 | ± 7.5 | ± 129.1 | ± 176.6 |
| n/n$_{tot}$ (n: number of analyses included weighted mean, n$_{tot}$: total number of analysis) | 22 / 67 | 11 / 83 | 4 / 67 | 19 / 73 | 4 / 62 | 4 / 75 | 9 / 64 | 3 / 75 |
| MSWD | 14.48 | 1.56 | 0.19 | 3.82 | 8.83 | 0.87 | 2.21 | 0.46 |





## 4 Discussion

During *in vacuo* stepwise crushing, the release of argon isotopes from the samples follows systematic patterns. The challenge is to link this release of argon from the samples to the different potential reservoirs of K and, as a next step, the geological meaning of the age and elemental ratios of K/Cl and Ca/Cl. Here, we first discuss potential issues related to the analytical quality of the data. Next, we discuss potential reservoirs of K and subsequently $^{40}Ar^*$ to link these options to our results, and to finally assess the ages and their broader implications.

### 4.1 Data quality

#### 4.1.1 Rursee 1a/1b BNV

We speculate that for the experiment Rursee 1a BNV, we sampled a smaller part of the argon reservoirs in the quartz minerals comparable to the first 10 % of the spectrum of Rursee 1b BNV. This is corroborated by the fact that for Rursee 1a BNV, 46 mg of quartz released 12.7 fA $^{39}Ar_K$ (0.3 fA/mg quartz), while for Rursee 1b BNV, 89.1 fA was released from 25 mg of quartz (3.6 fA/mg of quartz). We therefore do not further discuss the results of Rursee 1a BNV, but note that sample heterogeneity might also have contributed to this difference.

#### 4.1.2 Impact of blank correction

Blank correction procedure likely does not impact weighted mean age computation; however, it does influence the $^{40}Ar/^{36}Ar$ intercept of the inverse isochron. This is only the case when the regression line has a non-radiogenic intercept that is different from the atmospheric $^{36}Ar/^{40}Ar$. When the intercept is within the error overlapping with the atmospheric ratio, the blank correction only causes the point to move along the regression line as comes out of the discussion below as well. We described our blank correction procedure in methods (see supplementary file 1). The fact that we cannot mimic the dropping of the pestle when a sample is present in the tube provides limitations on how well we can determine the blank during the experiments. The blank tends to increase with higher number of pestle drops, but composition of this blank is atmospheric. For the test of the blank, we used quartz glass fragments to mimic zero-age minerals, as a blank determination using metal on metal impacts was considered to be an unrealistic scenario. As a next test we artificially increase the $^{40}Ar$ blank (and thus the $^{36}Ar$ blank) assuming atmospheric composition. If the data are located on the mixing line between radiogenic and atmospheric argon, this should not affect isochron age (pink part – final stage in fig. 5 or 11). We tested this for sample Rursee 1b BNV with an age of ~88 Ma. The $^{40}Ar/^{36}Ar$ intercepts increase with increasing blank values, and the weighted mean plateau ages change with a maximum of 2.5 Ma in the chosen example. We therefore conclude that the isotopic ages remain largely unaffected, by varying the amounts of atmospheric argon of the blanks. Note, that if the isochron is not a mixing line between radiogenic and atmospheric argon (e.g. blue and green parts in fig 5 or 11), this assumption is incorrect. The $^{40}Ar/^{36}Ar$ intercept is then pulled away from the real $^{40}Ar/^{36}Ar$ composition in the direction of the atmospheric $^{40}Ar/^{36}Ar$ intercept. Consequently, in the intercept with the inverse isochrons' X-axis (and thus age) will also be affected.





### 4.1.3 Recoil artefacts

These artefacts occur when $^{37}$Ar and $^{39}$Ar, which are formed from K and Ca isotopes, form with kinetic energy. As a result, they can travel from their original sites to other sites, potentially even into the adjacent phase (Turner & Cadogan, 1974; Foland, 1983; Lo & Onstott, 1989; Féraud & Courtillot, 1994; Baksi, 1994; Onstott et al., 1995; Villa, 1997). However, this phenomenon is assumed to have a smaller impact than that of the blank correction.

### 4.2 Potential reservoirs of K

To date, three main hypotheses are being debated as to the origin of the released argon in a stepwise crushing experiment. The first group (Qiu & Wijbrans, 2006, 2008; Bai et al., 2019) suggests that progressive crushing releases gases mainly from FIs and therefore represents FIs ages. The second group (e.g., Kendrick and Philips (2007)) discusses the possibility of K-bearing mineral inclusions within the inclusion cavity and/or in microcracks serving as argon reservoirs in the later stages of crushing. Obtained ages therefore represent mineral closure ages or a mixture of FIs and mineral ages.

In addition, the third potential source of potassium in the quartz minerals might be the presence of minor amounts of $K^+$ in the crystal lattice (Kendrick et al., 2011) of quartz minerals, which is representative of the formation age of veins. Hydrothermal quartz veins, characterised by their substitution in crystal structure, have been studied by Weil (1984) and Götze et al. (2021). These studies indicate that $Si^{4+}$ derived from hydrothermal quartz veins has the ability to be substituted by other ions such as $Al^{3+}$, $Ga^{3+}$, $Fe^{3+}$, $B^{3+}$, $Ge^{4+}$, $Ti^{4+}$, and $P^{4+}$. $Al^{3+}$ is most commonly replacing $Si^{4+}$ since it is found in significant quantities (~300-700 ppm) in quartz, based on EPMA data. Additionally, small quantities of monovalent ions such as $K^+$ may fill empty spaces in the crystal structure, serving as charge balancers for trivalent substitutional ions such as $Al^{3+}$ (Bambauer, 1961; Kats, 1962; Perny et al., 1992; Stalder et al., 2017; Potrafke et al., 2019). However, Jourdan et al. (2009) postulated that the substitution of these components may be so minor that it is even undetectable using a Secondary Ion Mass Spectrometer (SIMS). Furthermore, it is important to note that not all hydrothermal sources or quartz minerals have this particular form of substitution (Jourdan et al., 2009).

Apart from these potential $^{39}$Ar$_K$ reservoirs above, detrital minerals (e.g., mica present in the surrounding pelitic rock) that might be trapped by the quartz veins during the growth may also contribute to the obtained ages.

### 4.2.1 Identification of different K reservoirs in the Rursee quartz samples

During *in vacuo* stepwise crushing, the release of argon isotopes from the samples follows systematic patterns. Here, we attempt to link this release to the sequential contributions of different reservoirs of K and, thus, argon from the Rursee samples. The release patterns of $^{36}$Ar$_{air}$, $^{38}$Ar$_{Cl}$, $^{39}$Ar$_K$, and $^{40}$Ar* (Fig. 6) for all quartz vein samples may originate from multiple existing argon reservoirs.

Depending on the size (<10 μm), location, and generation of FIs, they may contribute successively to the argon release patterns in the early or middle stage of stepwise crushing. Figure 6 reveals that the concentration of $^{39}$Ar$_K$ increases throughout the



process of *in vacuo* stepwise crushing, while the concentration of other argon isotopes decreases. This suggests that K-
containing reservoirs were not opened in the first part of the experiment. The release patterns of $^{39}Ar_K$ can be categorised into
two distinct groups during stepwise crushing:

   a)   The *first group* of samples exhibits a small initial release during the early stages, followed by a decrease in the ~10th


step and an increase from the ~10th to ~35th step followed by a gradual decrease (Rursee 3 BPV, Rursee 5 BNV, and

Rursee 6 BPV).

b)   The *second group*, on the other hand, lacks the initial release of $^{39}Ar_K$ steps 1-10, but behaves the same for step 10

onwards with a gradual increase to the ~35th step followed by a gradual decrease (Rursee 1b BNV, Rursee 2 BPV,

Rursee 2.1 BNV, and Rursee 4 BPV).

The continuous rise in $^{39}Ar_K$ levels after ~10 steps in both sample groups, suggests that the gas release process can be divided
into at least two phases. Initially, during the first ~10 steps, $^{39}Ar_K$ is emitted from FIs in microcracks (secondary FIs). From
steps ~10th to ~70th, the release occurs as a result of mixing of potential pseudo-secondary FIs (~10- ~15th steps), mineral
inclusions and/or the crystal lattice of quartz veins. This interpretation is supported by the K/Cl correlation plots (Fig. 7), which
show a consistent lower K/Cl ratio until the ~10th step.
From the 10th to the 15th K/Cl ratio, it reaches ~1 with a steep rise for all quartz samples, and later (from ~20th step) this ratio
continues to increase steeply for *the second group* of samples, while it shows a less pronounced increase for *the first group* of
samples.
The lower K/Cl ratio may be attributed to the presence of Cl and a lack of or limited amounts of K in combination with
relatively constant low salinity levels (3.5-8 eq. wt.% NaCl) inside the FIs, which are likely to be opened in the early phase.
After most FIs have been mechanically opened, the subsequent rapid increase in K (reflected by the $^{39}Ar_K$) and the steady
decline in Cl (reflected by the $^{38}Ar_{Cl}$) occur throughout successive crushing steps and is reflected in the K/Cl ratio. Therefore,
this increase is most likely caused by the exhaustion of the Cl-rich FIs in combination with the presence of minerals containing
potassium and/or potassium from the crystal lattice of quartz that release their argon in the later crushing steps.
This approach to distinguish between FIs and other K reservoirs was first suggested by  (Kendrick et al., 2006, 2011): K/Cl
ratios ≤1 are representative for FIs and K/Cl ratios >1 for other sources. Therefore, if K/Cl ≤1, the obtained age corresponds
to the age of the FIs. If the K/Cl >1 the obtained age corresponds to the age of the trapped K-bearing mineral and/or K from
the crystal lattice (Kendrick et al., 2006, 2011). In our samples the K/Cl is greater than 1 after the first ~15±3 steps in all quartz
vein samples, indicating the presence of major K-related reservoir(s) other than FIs. It is worth noting that this is based on the
assumption that there are no other K-bearing phases, such as $KNO_3$, $K_2SO_4$ or $K_2CO_3$, rather than KCl dissolved in aqueous
FIs. This assumption seems to be verified by Raman analysis (see Figure A1), which does not show detectable peaks for these
alternative K-bearing phases. Therefore, K/Cl >1 suggests that K does not only relate to the salinity of the FIs, and at least one
major other source should be present, e.g., the crystal lattice of quartz and/or mineral inclusions in the quartz crystals and/or
in microcracks.





**Figure 7: K/Cl ratios plotted against analyzing steps for all quartz veins.**



### 4.2.2 K-bearing mineral inclusions

EPMA data (Table 2) from cleaned hand-picked fluid-rich separated quartz grains indicate the presence of sericite, chlorite-sericite and illite-sericite in the microfractures and in the cavities of fluid inclusion, which might explain the subsequent increase of $^{39}Ar_K$ from the ~10th analysing step onwards. The presence of these minerals (or mixtures) in the inclusion cavity and microfractures is also invisible under a binocular or petrographic microscope during the mineral separation, but it was also captured using electron-backscattered imaging (Fig. 8). In thin sections of quartz veins with associated host rock, illite-sericite and white mica are abundant in the surrounding pelitic layer of the Rursee formation (Fig. 9). These minerals that contain a significant amount of $K_2O$ are also detected by EPMA, in the separated quartz samples, especially in Rursee 2 BPV (see EPMA data, Table 2). High K concentrations (~8.8 wt. % $K_2O$) are likely related to intergrowth with sericite or a closely-related mineral.

Additionally, petrographic analysis of thin sections of whole rock samples representing both vein generations (BPV and BNV) show an abundance of chlorite in between the vein wall and host rock, as well as in fractures (Fig. 10). Despite the absence of K in the crystal structure of chlorite, traces of K were reported for chlorites in previous studies (Pacey et al., 2020; Li et al., 2022).







**Figure 8: Images of mineral inclusions under electron-backscattered SEM. Secondary minerals (e.g. chlorite, sericite and mica) occur in cavities and microfractures (pointed by white arrow) in separated fluid-rich quartz fraction as determined using EPMA.**

**Figure 9: Microscopic image of the quartz veins host rock matrix from the Rursee formation. (a) Cross polarizer (b) plane polarizer images of the pelitic host rock (Rursee 2 BPV). White arrows (image a) indicate the presence of the mica and sericite in the host pelitic rock. (c) Cross polarizer (d) plane polarizer images of the quartz veins matrix (Rursee 1 BNV). White arrow (image c) shows the presence of the quartz sub-grains. The presence of quartz sub-grains in the veins are due to the local tectonic activity, indicating that this period is correspond to tectonic activity.**







**Figure 10: Chloritization distribution in the vein wall and in fractures for both generations of quartz veins. (a) Plane (a.1) and cross (a.2) polarizer of bedding parallel veins: chloritization mainly between vein wall and host rock, and fractures. (b) Plane (b.1) and cross (b.2) polarizer of bedding normal veins: chloritization in fractures.**





### 4.2.3 K from crystal lattice and detrital minerals

EPMA analyses of quartz matrix indicate that K concentrations in the crystal lattice are below the detection limit of ~100 ppm.
A maximum K concentration of ~100 ppm K (for example, 100 ppm K in Rursee 2.1 BNV) and an age of 144 Ma would result
in ~16000 fA $^{40}Ar^*$ when measured on our Helix-MC mass spectrometer, which is a comparable amount of total $^{40}Ar^*$ released
from K-bearing mineral inclusion. Given the large amount of sample (~30 mg), this would translate into a significant
contribution of K from the crystal lattice of quartz. We therefore suggest that K in the crystal lattice may contribute to the
observed $^{40}Ar^*$ signals (see calculation on supplementary file 2).
In this study, argon molecules might also be derived from secondary minerals in cracks as well as embedded detrital minerals
(e.g., mica from host rock). This interpretation aligns with the observation that the homogenization temperatures of FIs within
the quartz veins are below the closure temperature for argon in detrital minerals. Under such conditions, the expected ages
from K-bearing detrital minerals would correspond to pre-Variscan periods, reflecting the age of the deposits hosting the quartz
veins, while the obtained ages are significantly younger in this study. Therefore, we infer that detrital minerals do not
significantly contribute to the $^{40}Ar^*$ signals.
To summarise, during the first stages (until the ~20$^{th}$ analysis steps) of the stepwise *vacuo* crushing, gases are likely released
only from FIs (secondary and pseudo-secondary, as is also observed for FI in garnets (Qiu & Wijbrans, 2006, 2008)). Huseynov
et al. (2024) demonstrated that a significant amount of fluid inclusion water can be extracted from these samples by a single
crushing step using a spindle crusher. In this study, throughout the crushing process, the total amount of argon released steadily
increases (Fig. 6). In the latter stages of the experiment (from the 20$^{th}$ analysing steps), the substantial release of $^{39}Ar_K$ isotopes
may support the hypothesis proposed by Kendrick and Philips (2007) and Kendrick et al., (2011), suggesting the presence of
K-bearing mineral inclusions in the samples and/or $^{40}Ar^*$ from the crystal lattice and also non-crushed small-sized FIs (<5 μm).
The presence of K-bearing mineral inclusions is also corroborated by EPMA data, and the presence of K in the lattice cannot
be ruled out for the Rursee samples.

### 4.3 Age spectra and isochrons

As aforementioned, the distribution of argon isotopes (Fig. 6) indicates that $^{39}Ar_K$ is derived from distinct sources, likely
mineral inclusions and/or eventually crystal lattice rather than FIs in particular in the later phase of the experiment, which was
used for the age determinations. These various sources of K, including fluid and mineral inclusions and/or crystal lattice, may
all contribute to the variability observed in the age spectra derived from the different samples. Due to the presence of $^{40}Ar_E$
from the FIs, the initial analytical stages of the analyses yield anomalously high ages in the first part of their age spectra (Fig.
4). Some samples (Rursee 3 BPV, Rursee 5 BNV, and Rursee 6 BPV) show a "pseudo-plateau" in the first part of the
experiment. The "pseudo-plateau" effect occurs between the 20-30$^{th}$ analysing steps, which may be associated with sudden
changes in K/Cl ratios (Fig. 7). These sudden changes may be due to sharp transition from fluids states reservoirs (e.g. small
sized FIs) to solid states reservoirs (e.g. K-bearing mineral inclusions). However, it does not occur in *the second group* quartz



samples (Rursee 1b BNV, Rursee 2 BPV, Rursee 4 BPV) revealing smooth transitions from fluids to solid states $^{39}Ar_K$
reservoirs. The transition for the Rursee 2.1 BNV is neither abrupt like for the *first group* samples nor smooth as for the *second*
*group* samples; hence, the impact of the "pseudo-plateau" is minimal.
The transition from fluid state reservoirs to solid state reservoirs can be supported by grain size distribution (see supplementary
file 4), indicating that fluid state reservoirs may remain unreleased beyond around 800 crushes (around the 20$^{th}$ analysis step).
However, the accumulation of small particles at the bottom of the crusher (non-recoverable size) after 800 crushes, may result
in the measured results not accurately representing the whole grain size distribution. As the grain size distribution depends on
many factors (i.e., crushing efficiency, presence of microcracks), even for separated clean quartz grains, that may be a factor
of difference for two groups.
The impact of $^{40}Ar_E$ results in inverse isochrons (Fig. 5) during the initial stage. The relationship between the $^{36}Ar/^{40}Ar$ and
$^{39}Ar/^{40}Ar$ for all samples resulted in a decrease in the $^{36}Ar/^{40}Ar$ ratio and an increase in the $^{39}Ar/^{40}Ar$ ratio (initial stage in Fig.
11). The presence of an elevated concentration of $^{36}Ar$ at the beginning of the experiment could be either due to the atmospheric
argon gas that is trapped in the stainless steel crusher and/or the original fragment surfaces and perhaps released during the
initial stage of crushing. Following the opening of FIs, the ratio of $^{36}Ar/^{40}Ar$ increases linearly with the ratio of $^{39}Ar/^{40}Ar$. This
is probably due to a decrease in excess argon throughout the crushing and an increase in $^{39}Ar_K$ associated with K-bearing
minerals and/or crystal lattice (intermediate stage in Fig. 11). In the last phase of $^{40}Ar/^{39}Ar$ analysis, the concentration of $^{39}Ar_K$
decreases (final stage in Fig. 11). This last part is particularly important for determining the age of quartz vein samples.
Inverse isochrons may assist in determining the age of FIs by linear regression of the data related to FIs. However, the high
amounts of excess argon in the system obscure geologically meaningful ages.





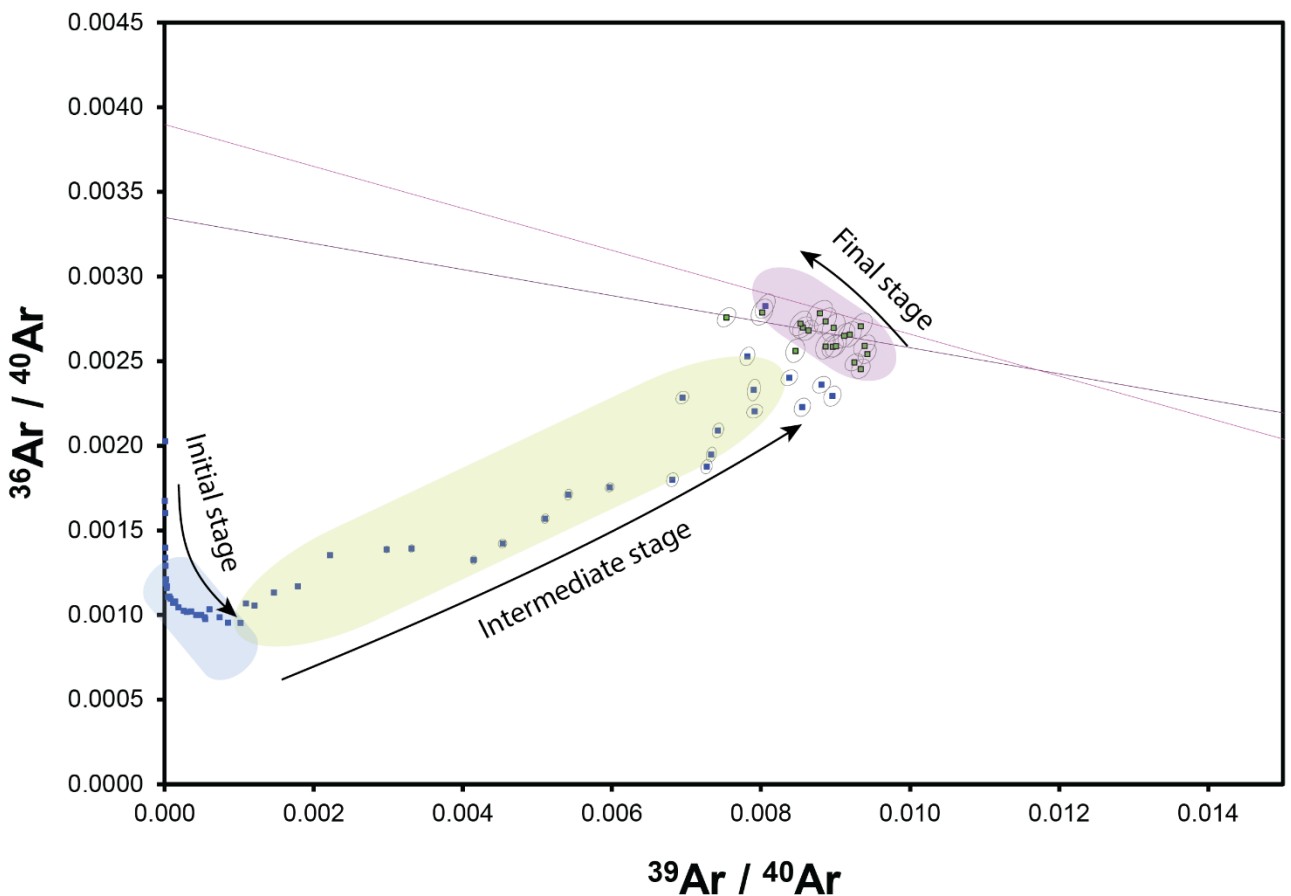

Figure 11: Inverse isochron representation of quartz veins (ex: Rursee 2.1 BNV): 3 stages: (1) initial stage with opening of fluid inclusions; (2) intermediate stage where argon is released from mineral inclusions or microfractures and/or crystal lattice; (3) Final stage of argon release from mineral inclusions and neglectable excess argon in samples.





**4.4 Implications**

Unlike studies that obtained consistent ages from FIs (Qiu & Wijbrans, 2006; Qiu et al., 2011; Bai et al., 2013, 2019), we were unable to date FIs in Rursee quartz samples, likely due to high $^{40}Ar_E$ concentrations and/or low salinity. While no age was determined for the FIs, $^{40}Ar/^{36}Ar$ ratios (above atmospheric but below 4000) indicate a mixed metamorphic-meteoric fluid source (Ballentine et al., 2002; Ozima & Podosek, 2002). Later, during the crushing experiment, the K-bearing mineral inclusions may provide geologically meaningful ages although the argon closure temperatures in quartz remain uncertain. For reference, the closure temperature of smaller size sericite grains (~20 μm) correspond to temperatures (~300-350 °C) (Glasmacher et al., 2001; Watson & Cherniak, 2003), while the vitrinite reflectance from psammatic and pelitic layers indicate maximum burial temperatures (220 ºC) near the Carboniferous-Permian boundary, with gradual cooling thereafter (Littke et al., 2012).

The obtained $^{40}Ar/^{39}Ar$ ages (117-84 Ma) differ from the interpretation based on structural analyses, which posit that veining occurred during the early Variscan Orogeny (Van Noten et al., 2007), possibly due to argon loss during cooling and/or recrystallization. The obtained ages may be influenced to some extent by the presence of neo-crystallized quartz sub-grains, although their volume appears relatively small (Fig. 9c). However, since the ages primarily reflect solid-phase reservoirs (i.e., K-bearing mineral inclusions) rather than fluid-phase components, it is likely that K-bearing solid-phase reservoirs intergrowth simultaneously with the recrystallization process.

Post-Variscan tectonic activity is known for southern Rhenish Massif due to late- and post-orogenic fault movements and coeval reactivation of Variscan structures leading to the fluids (re)activity (Herbst & Muller, 1969; Schwab, 1987; Korsch & Schäfer, 1991; Hein & Behr, 1994; Moe, 2000; Kirnbauer et al., 2012).

Given that reactivation of existing veins could have occurred without forming new fractures (Virgo et al., 2013), this reactivation is usually associated with the infiltration of high saline (>20 eq. wt.% NaCl) fluids in Central Europe and the Rhenish Massif (Behr et al., 1987; Redecke, 1992; Hein & Behr, 1994; Germann & Friedrich, 1999; Heijlen et al., 2001; Kučera et al., 2010).

This saline fluid activity is at odds with the low-salinity FIs (3.5-8 eq. wt.% NaCl) in the Rursee quartz veins (Van Noten et al., 2011). However they agree with low saline FIs in quartz veins of the Rhenish Massif, which are attributed to upward migration of Variscan fluid remnants during Jurassic-Cretaceous reactivation (Kirnbauer et al., 2012).

Near Rursee (Stavelot Inlier), low saline (0.2-7.2 eq. wt.% NaCl) and high-temperature fluid activity (~250 °C) along the Variscan front reflect warm meteoric fluid circulation (Schroyen & Muchez, 2000). Such warm, low saline fluids may have also contributed to chloritization of veins in the in Rursee outcrops. We propose that tectonic activity and quartz vein reactivation could possibly explain the observed $^{40}Ar/^{39}Ar$ ages, as low saline Variscan fluids perhaps moved along the reactivated fractures, forming new quartz minerals within the Variscan-related veins during Jurassic-Cretaceous tectonic activity (i.e. opening of North Atlantic).



**Table 2 Elemental analysis of quartz grain, microcracks, and mineral inclusions in quartz vein samples under EPMA.**

EPMA analysis of mineral inclusions and microfractures of clean fraction of quartz veins grain of Rursee samples (wt.%).

| | Sample ID | Grain ID | SiO$_2$ | TiO$_2$ | Al$_2$O$_3$ | FeO | MnO | MgO | CaO | Na$_2$O | K$_2$O | P$_2$O$_5$ | Cl | F | BaO | ZrO$_2$ | O | H2O | TOTAL |
|---|---|---|---|---|---|---|---|---|---|---|---|---|---|---|---|---|---|---|---|
| Internal standard of UU | KL2-1 | | 51.90 | 2.56 | 13.50 | 10.74 | 0.17 | 7.47 | 10.74 | 2.39 | 0.48 | 0.23 | bdl | bdl | bdl | bdl | 0.00 | 0.00 | 100.01 |
| | | | 51.20 | 2.51 | 13.36 | 10.79 | 0.17 | 7.46 | 10.83 | 2.38 | 0.47 | 0.27 | 0.00 | bdl | bdl | bdl | 0.00 | 0.00 | 99.28 |
| | | | 50.80 | 2.55 | 13.41 | 10.98 | 0.16 | 7.31 | 10.96 | 2.34 | 0.51 | 0.26 | 0.00 | bdl | bdl | 0.00 | 0.00 | 0.00 | 99.13 |
| Internal standard of UU | ATHO-1 | | 75.73 | 0.21 | 12.23 | 3.28 | 0.11 | 0.11 | 1.67 | 3.85 | 2.73 | 0.02 | 0.05 | 0.06 | 0.09 | bdl | 0.00 | 0.00 | 100.10 |
| | | | 75.61 | 0.24 | 12.44 | 3.47 | 0.12 | 0.13 | 1.63 | 3.73 | 2.78 | 0.05 | 0.05 | 0.03 | 0.00 | 0.02 | 0.00 | 0.00 | 100.29 |
| | | | 75.69 | 0.28 | 12.30 | 3.43 | 0.12 | 0.11 | 1.60 | 3.78 | 2.81 | 0.02 | 0.03 | 0.04 | 0.04 | 0.10 | 0.00 | 0.00 | 100.35 |
| Rursee quartz veins | Rursee 1 BNV | B_1.01a | 99.75 | 0.02 | 0.00 | 0.00 | 0.00 | bdl | 0.01 | 0.02 | 1.00 | 0.01 | 0.59 | bdl | bdl | bdl | 0.00 | 0.00 | 101.21 |
| | | B_1.02a | 55.07 | bdl | 0.31 | 0.29 | 0.01 | 0.61 | 0.88 | 0.39 | 0.51 | 0.13 | 0.40 | bdl | 0.03 | 0.01 | 0.00 | 0.00 | 58.57 |
| | | B_1.03a | 71.35 | 0.02 | 0.69 | 2.83 | 0.07 | 14.04 | 0.15 | 0.38 | 0.51 | 0.05 | 0.12 | bdl | bdl | 0.00 | 0.00 | 0.00 | 90.05 |
| | | B_1.03b | 38.76 | 0.04 | 2.80 | 1.85 | 0.02 | 6.30 | 0.74 | 1.53 | 0.95 | 0.13 | 0.30 | bdl | 0.04 | bdl | 0.00 | 0.00 | 53.29 |
| | | B_1.06a | 47.68 | 0.01 | 37.73 | 0.04 | bdl | 0.04 | 0.19 | 5.61 | 0.86 | 0.05 | 0.06 | 0.04 | bdl | bdl | 0.00 | 0.00 | 92.27 |
| | | B_1.06b | 99.52 | bdl | 0.67 | 0.01 | 0.01 | 0.00 | 0.05 | 0.04 | 0.04 | 0.05 | 0.00 | 0.00 | 0.02 | 0.05 | 0.00 | 0.00 | 100.45 |
| | | B_1.06c | 92.43 | 0.02 | 5.79 | bdl | bdl | 0.04 | 0.05 | 1.01 | 0.28 | 0.01 | 0.03 | bdl | 0.05 | bdl | 0.00 | 0.00 | 99.60 |
| | | B_1.06d | 50.05 | 0.03 | 25.07 | 0.01 | 0.02 | 0.10 | 0.08 | 2.44 | 3.24 | 0.05 | 0.03 | 0.02 | 0.00 | bdl | 0.00 | 0.00 | 81.10 |
| | | B_1.10a | 50.90 | 0.03 | 33.68 | 0.13 | bdl | 0.13 | 0.05 | 1.66 | 6.22 | 0.00 | 0.12 | 0.04 | 0.04 | bdl | 0.00 | 0.00 | 92.98 |
| | | B_1.11a | 52.33 | bdl | 31.93 | 0.09 | 0.00 | 0.08 | 0.07 | 0.29 | 8.08 | 0.07 | 0.03 | 0.02 | 0.15 | bdl | 0.00 | 0.00 | 93.00 |
| | | B_1.15a | 68.06 | 0.02 | 24.01 | 0.00 | 0.00 | 0.10 | 0.39 | 3.14 | 0.93 | bdl | 0.13 | bdl | 0.03 | 0.03 | 0.00 | 0.00 | 96.82 |
| | Rursee 2 BPV | B_2.02a | 78.24 | bdl | 0.25 | 0.32 | 0.00 | 0.04 | 0.37 | 0.22 | 0.03 | 0.19 | 0.05 | bdl | 0.05 | 0.03 | 0.00 | 0.00 | 79.76 |
| | | B_2.02b | 13.65 | 0.03 | 1.51 | 59.78 | bdl | 0.31 | 0.37 | 1.52 | 0.95 | 0.16 | 0.61 | 0.02 | bdl | 0.04 | 0.00 | 0.00 | 78.90 |
| | | B_2.03a | 47.72 | 0.07 | 6.23 | 5.85 | 0.15 | 5.35 | 3.09 | 1.64 | 0.40 | 0.05 | 0.20 | 0.04 | bdl | 0.01 | 0.00 | 0.00 | 70.79 |
| | | B_2.03b | 55.38 | 0.58 | 2.11 | 12.35 | 0.34 | 9.12 | 12.46 | 0.64 | 0.24 | 0.02 | 0.07 | bdl | 0.04 | bdl | 0.00 | 0.00 | 93.29 |
| | | B_2.03c | 35.63 | bdl | 0.79 | 37.24 | 0.15 | 0.22 | 1.06 | 1.02 | 0.56 | 0.12 | 0.25 | 0.00 | bdl | 0.01 | 0.00 | 0.00 | 77.02 |
| | | B_2.04a | 24.10 | 0.01 | 22.59 | 30.81 | 0.18 | 7.20 | 0.03 | 0.04 | 0.03 | 0.05 | 0.01 | 0.00 | 0.03 | 0.01 | 0.00 | 0.00 | 85.09 |
| | | B_2.05a | 24.15 | 0.04 | 22.79 | 31.10 | 0.15 | 8.04 | bdl | 0.01 | bdl | 0.00 | 0.00 | bdl | bdl | bdl | 0.00 | 0.00 | 86.14 |
| | | B_2.09a | 23.19 | 0.04 | 23.09 | 33.29 | 0.20 | 7.22 | 0.03 | 0.01 | 0.03 | 0.00 | 0.02 | 0.02 | 0.00 | 0.01 | 0.00 | 11.05 | 98.18 |
| | | B_2.09b | 47.70 | 0.05 | 35.29 | 0.83 | 0.02 | 0.88 | 0.01 | 0.28 | 8.78 | 0.00 | 0.02 | 0.13 | 0.13 | bdl | 0.00 | 4.59 | 98.70 |
| | | B_2.12a | 100.15 | bdl | 0.06 | 1.62 | 0.00 | 0.01 | 0.02 | 0.01 | 0.00 | 0.06 | 0.02 | bdl | bdl | bdl | 0.00 | 0.00 | 101.78 |
| | | B_2.12b | 56.62 | bdl | 1.83 | 26.39 | 0.02 | 0.04 | 0.05 | 0.13 | 0.33 | 0.55 | 0.32 | 0.07 | 0.00 | 0.00 | 0.00 | 0.00 | 86.34 |




**Table 2 (continue).**

| Sample ID | | Grain ID | SiO$_2$ | TiO$_2$ | Al$_2$O$_3$ | FeO | MnO | MgO | CaO | Na$_2$O | K$_2$O | P$_2$O$_5$ | Cl | F | BaO | ZrO$_2$ | O | H2O | TOTAL |
|---|---|---|---|---|---|---|---|---|---|---|---|---|---|---|---|---|---|---|---|
| Rursee quartz veins | Rursee 2.1 BNV | B_2.1-03 | 97.91 | 0.02 | 0.04 | 1.69 | 0.00 | 0.01 | 0.00 | 0.01 | 0.00 | 0.00 | 0.02 | bdl | 0.00 | bdl | 0.00 | 0.00 | 99.58 |
| | | B_2.1-05a | 100.31 | bdl | 0.32 | 0.19 | 0.02 | 0.02 | 0.00 | 0.02 | 0.01 | 0.00 | bdl | bdl | bdl | bdl | 0.00 | 0.00 | 100.80 |
| | | B_2.1-07a | 45.73 | bdl | 0.36 | 0.11 | 29.58 | 0.22 | 1.79 | 0.38 | 1.25 | 0.02 | 0.29 | bdl | bdl | bdl | 0.00 | 0.00 | 77.74 |
| | | B_2.1-07b | 52.22 | 0.02 | 0.24 | 0.05 | 24.09 | 0.21 | 1.52 | 0.29 | 0.89 | 0.00 | 0.22 | bdl | 0.03 | bdl | 0.00 | 0.00 | 78.46 |
| | | B_2.1-07c | 61.70 | 0.01 | 0.23 | 0.10 | 21.09 | 0.20 | 1.21 | 0.18 | 0.72 | 0.01 | 0.25 | bdl | 0.02 | bdl | 0.00 | 0.00 | 84.39 |
| | | B_2.1-11a | 40.79 | 0.07 | 26.15 | 19.64 | 0.10 | 3.03 | 0.01 | 0.28 | 3.65 | 0.03 | 0.03 | 0.05 | 0.12 | 0.05 | 0.00 | 0.00 | 94.03 |
| | | B_2.1-11b | 67.46 | 0.01 | 15.25 | 4.91 | 0.01 | 0.84 | 0.07 | 0.14 | 3.41 | 2.12 | 0.03 | 0.13 | bdl | bdl | 0.00 | 0.00 | 94.24 |
| | Rursee 3 BPV | B_3.02a | 42.77 | 0.01 | 15.84 | 22.60 | 0.17 | 4.89 | 0.07 | 0.02 | 0.67 | 0.07 | 0.15 | bdl | bdl | bdl | 0.00 | 0.00 | 87.12 |
| | | B_3.02b | 66.75 | 0.02 | 9.82 | 13.59 | 0.08 | 3.14 | bdl | 0.05 | 0.17 | 0.02 | 0.03 | 0.00 | 0.04 | 0.02 | 0.00 | 0.00 | 93.70 |
| | | B_3.02c | 90.77 | 0.02 | 3.20 | 3.97 | 0.03 | 1.03 | bdl | bdl | 0.04 | 0.01 | 0.00 | 0.00 | 0.00 | 0.05 | 0.00 | 0.00 | 99.08 |
| | | B_3.02d | 24.66 | 0.01 | 21.55 | 27.73 | 0.16 | 6.22 | 0.06 | 0.04 | 0.79 | 0.53 | 0.21 | 0.07 | bdl | 0.00 | 0.00 | 0.00 | 81.96 |
| | | B_3.05a | 30.48 | 0.03 | 23.88 | 25.27 | 0.16 | 7.25 | 0.03 | 0.03 | 0.29 | 0.04 | 0.06 | bdl | bdl | bdl | 0.00 | 0.00 | 87.44 |
| | | B_3.05b | 39.46 | 0.00 | 30.20 | 13.10 | 0.06 | 2.88 | 0.05 | 0.16 | 2.88 | 0.18 | 0.10 | 0.08 | bdl | 0.09 | 0.00 | 0.00 | 89.24 |
| | | B_3.05c | 64.23 | 0.03 | 16.31 | 2.75 | 0.00 | 1.11 | 0.01 | 0.11 | 4.01 | 0.01 | 0.06 | bdl | bdl | bdl | 0.00 | 0.00 | 88.58 |
| | | B_3.08a | 56.03 | 0.26 | 21.87 | 6.45 | 0.01 | 2.10 | 0.02 | 0.15 | 4.74 | 0.03 | 0.00 | 0.09 | 0.05 | bdl | 0.00 | 0.00 | 91.79 |
| | | B_3.08b | 70.37 | 0.01 | 10.47 | 12.40 | 0.05 | 2.86 | 0.03 | 0.00 | 0.62 | 0.01 | 0.02 | 0.00 | 0.03 | bdl | 0.00 | 0.00 | 96.82 |
| | | B_3.08c | 71.23 | bdl | 15.55 | 3.49 | 0.03 | 0.74 | 0.04 | 7.15 | 0.08 | bdl | 0.01 | 0.01 | bdl | bdl | 0.00 | 0.00 | 98.29 |
| | | B_3.08d | 32.87 | 0.73 | 19.01 | 25.07 | 0.13 | 8.40 | 0.01 | 0.07 | 0.09 | 0.01 | 0.01 | bdl | bdl | 0.01 | 0.00 | 0.00 | 86.40 |
| | | B_3.11a | 64.26 | 0.01 | 1.66 | 9.68 | 0.06 | 0.00 | 0.23 | 0.33 | 0.13 | 0.17 | 0.16 | 0.02 | 0.03 | bdl | 0.00 | 0.00 | 76.76 |
| | | B_3.11b | 51.24 | bdl | 12.87 | 20.70 | 0.11 | 4.48 | 0.01 | bdl | 0.01 | bdl | 0.01 | 0.02 | 0.03 | 0.05 | 0.00 | 0.00 | 89.48 |
| | Rursee 4 BPV | B_4.02a | 47.91 | 0.07 | 3.98 | 1.07 | 0.01 | 0.22 | 0.58 | 2.30 | 2.47 | 0.15 | 0.34 | 0.10 | 0.06 | bdl | 0.00 | 0.00 | 59.23 |
| | | B_4.03 | 100.84 | 0.00 | bdl | 0.02 | 0.00 | 0.00 | 0.02 | 0.00 | 0.02 | bdl | 0.02 | bdl | 0.08 | bdl | 0.00 | 0.00 | 100.78 |
| | | B_4.04 | 32.13 | 0.05 | 3.95 | 22.69 | 0.00 | 0.15 | 6.00 | 0.63 | 0.64 | 0.17 | 0.27 | 0.13 | bdl | 0.01 | 0.00 | 0.00 | 66.80 |
| | | B_4.05 | 85.80 | 0.00 | 0.03 | 0.01 | 0.02 | 0.02 | 0.32 | 0.02 | 0.03 | 0.24 | 0.03 | bdl | 0.71 | bdl | 0.00 | 0.00 | 87.15 |
| | | B_4.08 | 78.34 | 0.01 | 0.73 | 0.04 | 10.49 | 0.22 | 0.96 | 0.07 | 0.09 | 0.00 | 0.06 | bdl | bdl | bdl | 0.00 | 0.00 | 90.27 |
| | Rursee 5 BNV | B_5-03 | 60.18 | 0.12 | 1.28 | 1.08 | 0.05 | 1.33 | 3.01 | 0.98 | 0.97 | 0.11 | 0.37 | 0.00 | bdl | bdl | 0.00 | 0.00 | 69.38 |
| | | B_5-05 | 96.03 | 0.12 | 0.42 | 0.22 | 0.03 | 0.02 | 0.08 | 0.07 | 0.11 | 0.04 | 0.01 | bdl | 0.00 | 0.00 | 0.00 | 0.00 | 97.09 |
| | | B_5-06a | 39.40 | 0.03 | 5.43 | 49.86 | 0.21 | 0.23 | 0.17 | 0.27 | 0.51 | 0.93 | 0.19 | 0.00 | bdl | 0.07 | 0.00 | 0.00 | 97.27 |
| | | B_5-06b | 48.32 | 0.09 | 36.38 | 0.65 | 0.00 | 0.66 | 0.03 | 0.45 | 9.16 | 0.01 | 0.01 | 0.10 | 0.15 | bdl | 0.00 | 0.00 | 95.97 |
| | | B_5-06c | 47.07 | 0.05 | 34.95 | 0.69 | bdl | 0.67 | 0.06 | 0.35 | 9.02 | 0.00 | 0.02 | 0.14 | 0.06 | 0.06 | 0.00 | 0.00 | 93.12 |
| | | B_5-07 | 83.67 | 0.01 | 8.19 | 0.07 | bdl | 0.03 | 0.13 | 0.09 | 2.24 | 0.07 | 0.03 | bdl | bdl | 0.00 | 0.00 | 0.00 | 94.48 |




**Table 2 (continue).**

| | Sample ID | Grain ID | SiO$_2$ | TiO$_2$ | Al$_2$O$_3$ | FeO | MnO | MgO | CaO | Na$_2$O | K$_2$O | P$_2$O$_5$ | Cl | F | BaO | ZrO$_2$ | O | H$_2$O | TOTAL |
|---|---|---|---|---|---|---|---|---|---|---|---|---|---|---|---|---|---|---|---|
| Rursee quartz veins | Rursee 6 BPV | B_6-07b | 85.19 | 0.02 | 8.13 | 0.00 | bdl | bdl | 0.12 | 1.28 | 0.71 | 0.07 | 0.03 | bdl | bdl | 0.02 | 0.00 | 0.00 | 95.52 |
| | | B_6-07a | 79.39 | 0.00 | 3.32 | 0.00 | 0.01 | 0.03 | 0.31 | 0.06 | 0.94 | 0.24 | 0.09 | bdl | bdl | bdl | 0.00 | 0.00 | 84.32 |
| | | B_6-16a | 49.44 | bdl | 37.34 | 0.06 | bdl | 0.10 | 0.08 | 5.13 | 1.63 | 0.01 | 0.02 | 0.05 | 0.04 | bdl | 0.00 | 0.00 | 93.85 |
| | | B_6-16b | 57.16 | 0.01 | 19.71 | 0.03 | bdl | 0.06 | 0.16 | 2.85 | 1.57 | 0.03 | 0.03 | bdl | 0.05 | bdl | 0.00 | 0.00 | 81.61 |
| | | B_6-17 | 30.47 | 0.03 | 21.91 | 31.64 | 0.17 | 6.42 | 0.01 | 0.02 | bdl | 0.02 | 0.02 | bdl | bdl | bdl | 0.00 | 0.00 | 90.55 |
| | | B_6-19 | 69.23 | 0.04 | 17.80 | 0.12 | bdl | 0.04 | 0.04 | 0.46 | 4.81 | 0.00 | 0.03 | 0.00 | 0.01 | bdl | 0.00 | 0.00 | 92.51 |
| Internal standard of UU | KL2-2 | | 50.09 | 2.55 | 13.10 | 10.78 | 0.17 | 6.99 | 10.80 | 2.33 | 0.47 | 0.25 | 0.00 | 0.00 | 0.01 | bdl | 0.00 | 0.00 | 97.49 |
| | | | 50.37 | 2.59 | 13.11 | 10.79 | 0.14 | 6.93 | 10.94 | 2.42 | 0.51 | 0.25 | 0.00 | bdl | 0.04 | bdl | 0.00 | 0.00 | 98.00 |
| | | | 50.27 | 2.65 | 13.13 | 10.83 | 0.16 | 6.97 | 10.93 | 2.30 | 0.46 | 0.27 | 0.01 | bdl | bdl | 0.01 | 0.00 | 0.00 | 97.95 |
| Internal standard of UU | ATHO-2 | | 74.10 | 0.23 | 12.11 | 3.37 | 0.10 | 0.11 | 1.63 | 3.60 | 2.80 | 0.03 | 0.05 | 0.04 | 0.04 | 0.02 | 0.00 | 0.00 | 98.23 |
| | | | 74.00 | 0.24 | 12.22 | 3.38 | 0.10 | 0.09 | 1.59 | 3.60 | 2.68 | 0.03 | 0.04 | 0.10 | 0.06 | 0.01 | 0.00 | 0.00 | 98.13 |
| | | | 74.39 | 0.21 | 12.11 | 3.41 | 0.10 | 0.09 | 1.63 | 3.55 | 2.70 | 0.06 | 0.04 | 0.02 | 0.08 | 0.08 | 0.00 | 0.00 | 98.48 |
| Internal standard of UU | KL2-3 | | 51.64 | 2.60 | 13.46 | 10.89 | 0.19 | 7.46 | 10.98 | 2.33 | 0.47 | 0.30 | 0.00 | bdl | 0.00 | 0.02 | 0.00 | 0.00 | 100.29 |
| | | | 50.51 | 2.62 | 13.24 | 10.82 | 0.13 | 7.29 | 10.81 | 2.28 | 0.48 | 0.29 | 0.00 | bdl | bdl | 0.04 | 0.00 | 0.00 | 98.43 |
| | | | 50.83 | 2.61 | 13.13 | 10.97 | 0.15 | 7.29 | 11.01 | 2.30 | 0.48 | 0.29 | 0.01 | bdl | bdl | 0.02 | 0.00 | 0.00 | 99.00 |
| Internal standard of UU | ATHO-3 | | 76.25 | 0.24 | 12.22 | 3.29 | 0.10 | 0.09 | 1.62 | 3.65 | 2.77 | 0.04 | 0.05 | 0.06 | 0.03 | 0.11 | 0.00 | 0.00 | 100.53 |
| | | | 75.81 | 0.24 | 12.01 | 3.32 | 0.10 | 0.10 | 1.62 | 3.55 | 2.76 | 0.02 | 0.03 | 0.04 | 0.08 | bdl | 0.00 | 0.00 | 99.63 |
| | | | 75.50 | 0.26 | 12.08 | 3.39 | 0.10 | 0.11 | 1.63 | 3.69 | 2.68 | 0.00 | 0.03 | 0.08 | 0.05 | 0.07 | 0.00 | 0.00 | 99.67 |

EPMA analysis of crystal lattice of clean fraction of quartz veins grain of Rursee samples (wt.%).

| | Veins generation | Grain ID | SiO$_2$ | TiO$_2$ | Al$_2$O$_3$ | FeO | MnO | MgO | CaO | Na$_2$O | K$_2$O | P$_2$O$_5$ | Cl | F | BaO | ZrO$_2$ | O | H$_2$O | TOTAL |
|---|---|---|---|---|---|---|---|---|---|---|---|---|---|---|---|---|---|---|---|
| Rursee quartz veins | Bedding Normal Veins | AH2.1_04 | 101.97 | bdl | 0.03 | 0.00 | bdl | 0.00 | bdl | 0.00 | 0.00 | bdl | 0.00 | 0.01 | 0.00 | bdl | 0.00 | | 101.91 |
| | | AH2.1_05 | 101.47 | 0.00 | bdl | bdl | 0.00 | bdl | 0.00 | 0.01 | bdl | 0.00 | 0.00 | bdl | 0.01 | 0.02 | 0.00 | | 101.38 |
| | | AH2.1_07 | 101.68 | 0.00 | 0.03 | bdl | 0.00 | 0.01 | bdl | 0.01 | 0.01 | 0.00 | 0.01 | bdl | 0.02 | bdl | 0.00 | | 101.68 |
| | | AH2.1_09 | 102.02 | bdl | bdl | bdl | 0.01 | 0.01 | bdl | bdl | 0.01 | 0.00 | bdl | bdl | 0.00 | bdl | 0.00 | | 101.89 |
| | Bedding Parallel Veins | AH2_05 | 102.67 | bdl | 0.01 | bdl | bdl | 0.00 | 0.01 | bdl | 0.00 | bdl | 0.00 | bdl | 0.01 | bdl | 0.00 | | 102.61 |
| | | AH2_06 | 102.41 | 0.02 | 0.08 | bdl | bdl | bdl | 0.00 | 0.02 | 0.01 | 0.00 | 0.00 | bdl | 0.06 | bdl | 0.00 | | 102.48 |
| | | AH2_07 | 101.69 | bdl | 0.05 | 0.00 | bdl | 0.01 | bdl | bdl | bdl | 0.01 | 0.00 | bdl | 0.00 | 0.02 | 0.00 | | 101.61 |
| | | AH2_08 | 101.42 | 0.01 | 0.03 | 0.01 | 0.00 | bdl | 0.00 | 0.00 | bdl | 0.01 | 0.01 | bdl | bdl | 0.07 | 0.00 | | 101.47 |
| | | AH2_09 | 102.34 | 0.01 | 0.04 | bdl | 0.00 | bdl | bdl | bdl | bdl | bdl | 0.00 | bdl | 0.03 | bdl | 0.00 | | 102.22 |

* bdl - below detection limit



## 5 Conclusions

- The analysis of argon isotope patterns and their interpretations (including K/Cl and inverse isochrons) indicate that the main reservoir $^{39}Ar_K$ for geologically meaningful ages originated from the K-bearing minerals (illite-sericite and some possible chlorite) in quartz vein microcracks and/or inclusions cavities and/or crystal lattice of quartz.

- The determination of a primary crystallization age of the quartz veins is not feasible owing to the low amount of K in FIs and the high amount of excess argon within the FIAs resulting in anomalously old apparent ages in the first ~20th crushing steps.

- Estimates for a closure temperature for argon in K-bearing minerals inclusions is higher than the homogenization temperature of FIs of quartz. However, the activity of argon diffusion within the crystal lattice of quartz is high at this temperature. This may reveal that the apparent ages obtained belong to the moment of reactivation-recrystallisation of veins or its cooling moment.

- The ages obtained from the quartz samples span the Jurassic-Cretaceous period. The presence of neo-crystallized quartz sub-grains in the veins is due to the local tectonic activity, indicating that this period is corresponds to tectonic activity of the Rhenish massif.

**Data availability**

All data is included in the text and/or supplementary files.

**Author contributions**

The manuscript was primarily authored by Akbar Aydin Oglu Huseynov, the corresponding author. Co-authors Prof. Dr. Klaudia F. Kuiper, Em. Prof. Dr. Jan R. Wijbrans, and Dr. Jeroen van der Lubbe made substantial contributions to data interpretation and refinement of the manuscript, significantly enhancing its clarity and depth.

**Competing interest**

The authors declare that they have no conflict of interest.

**Acknowledgement**

This study has been funded by the FluidNET Consortium of EU H2020 Marie Skłodowska-Curie Action (No. 956127). We would like to thank Stefan Groen for his assistance in data management and the $^{40}Ar/^{39}Ar$ analysis laboratory, as well as Roel van Elsas for his help in the Mineral Separation Laboratory at VU Amsterdam. We would like to express our gratitude to



Bouke Lacet for preparing the epoxy grain mounts and thin sections and to Eric Hellebrand and Tilly Bouten for the EPMA
analysis. Finally, this study would have been impossible without the support of Prof. Dr. Janos L. Urai, who introduced us to
the Rursee outcrops, who unfortunately passed away.

**Supplementary file 1** Impact of blank correction on age.
**Supplementary file 2** K (and thus $^{40}Ar^*$) contribution from chlorite is estimated from 2D electron backscattered images.
**Supplementary file 3** Analytical data that have been used for $^{40}Ar/^{39}Ar$ dating.
**Supplementary file 4** Grain size distribution analysis of separated fluid-rich quartz fraction after crushing.

**Figure A1:** Raman spectroscopy of fluid inclusion from Rursee quartz veins. **(a)** Microscopic image of an epoxied and polished
fluid-rich quartz fraction. The fluid inclusion that underwent Raman analysis is represented by the red box. **(b)** The Raman
plot is presented with the wavelength on the x-axis and intensity on the y-axis. The Raman spectra shows a stretching band in
the wavelength range of 3000 to 3700 cm$^{-1}$, which indicates the presence of an aqueous system.
**Figure B1:** Normal isochron plots of all quartz veins samples.
**Table C1:** Rursee quartz veins samples J values and MDF with analytical error.




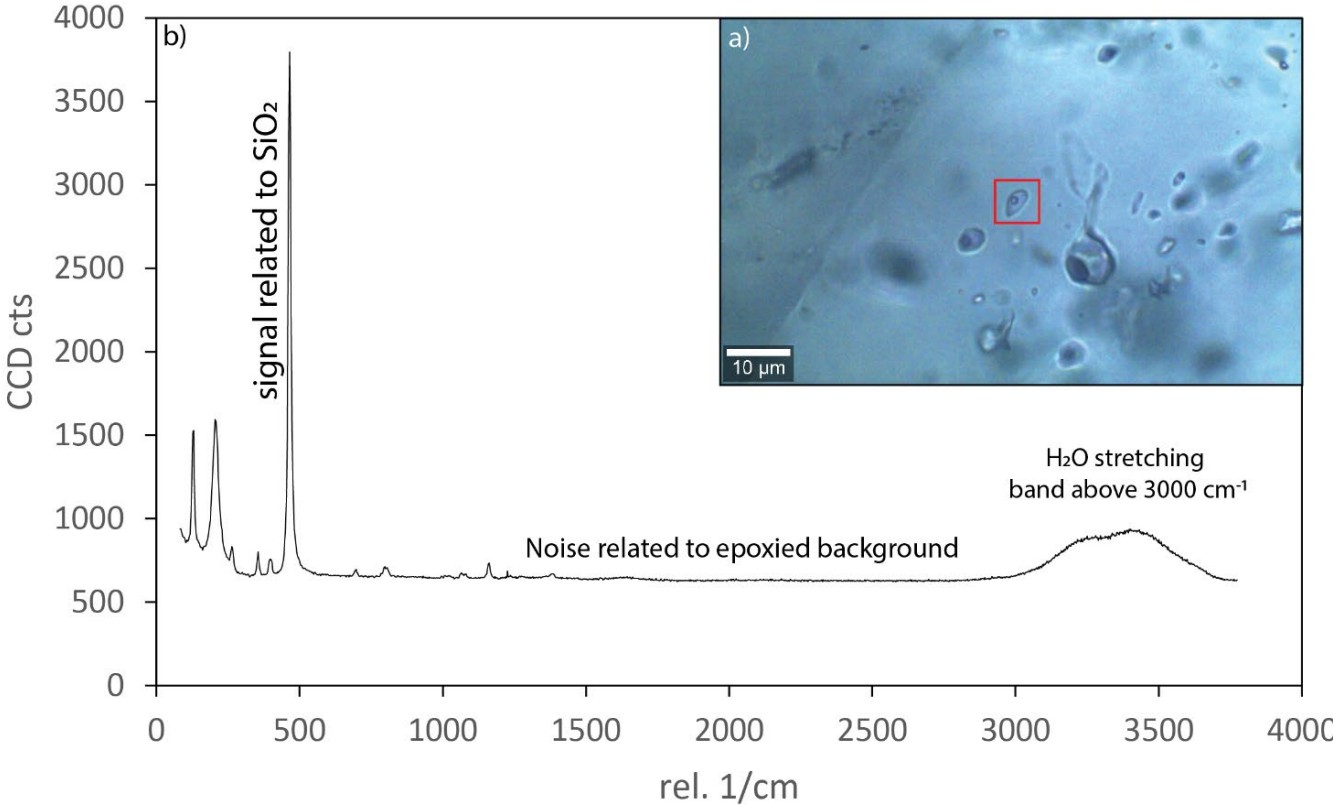

**Figure A1: Raman spectroscopy of fluid inclusion from Rursee quartz veins. (a) Microscopic image of an epoxied and polished fluid-**
**rich quartz fraction. The fluid inclusion that underwent Raman analysis is represented by the red box. (b) The Raman plot is**
**presented with the wavelength on the x-axis and intensity on the y-axis. The Raman spectra shows a stretching band in the**
**wavelength range of 3000 to 3700 cm$^{-1}$, which indicates the presence of an aqueous system.**






**Figure B1: Normal isochron plots of all quartz veins samples.**





**Table C1: Rursee quartz veins samples J values and MDF with analytical error.**

| Sample ID | Sample ID Ar | MDF | 1σ % | J - value | 1σ % |
|---|---|---|---|---|---|
| **Rursee 1a BNV** | R01a | 0.99635 | ± 0.04 | 0.0034347 | ± 0.06 |
| **Rursee 1b BNV** | R01b | 0.99519 | ± 0.04 | 0.0034737 | ± 0.06 |
| **Rursee 2 BPV** | R02 | 0.99469 | ± 0.03 | 0.0035113 | ± 0.03 |
| **Rursee 2.1 BNV** | R021 | 0.99492 | ± 0.03 | 0.0034868 | ± 0.04 |
| **Rursee 3 BPV** | R03 | 0.99868 | ± 0.03 | 0.0035113 | ± 0.03 |
| **Rursee 4 BPV** | R04 | 0.99749 | ± 0.03 | 0.0035113 | ± 0.03 |
| **Rursee 5 BNV** | R05 | 0.99494 | ± 0.04 | 0.0034868 | ± 0.04 |
| **Rursee 6 BPV** | R06 | 0.99709 | ± 0.04 | 0.0034868 | ± 0.04 |





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
