# Peer review of "40Ar/39Ar age constraints on the formation of fluid-rich quartz veins from the NW Rhenohercynian zone (Rursee area, Germany)"

_Geochronology, 2024_

## Author Response (AR1)

**$^{40}$Ar/$^{39}$Ar age constraints on the formation of fluid-rich quartz veins from the NW Rhenohercynian zone (Rursee area, Germany)**

Akbar A.O. Huseynov, Jan R. Wijbrans, Klaudia F. Kuiper, and Jeroen van der Lubbe

**Authors response to Referee 1:**

- **Dear Referee,**

**Thank you for your contribution on our submitted manuscript, we appreciate deeply your comments and consider your valuable thoughts on our revised version of manuscript.**

*I would like to start my comments by saying that I have never dated fluid and mineral inclusions in quartz by 40Ar/39Ar or any other method myself. However, I have experience in 40Ar/39Ar dating, including experience in dating mineral inclusions in pyrite, and I am also quite familiar with the U,Th-He method of dating mineral inclusions in sulfides. Of course, different approaches are used in all of the above cases, but I think I am qualified to evaluate the paper. Speaking directly about the paper, I read it with great interest. The article is impeccably written and the research itself is technically sound with ample material. On the whole, I have no complaints about this part of the paper, except perhaps that the colours of the regression lines in Figures 5, 11 and B1 are not contrasted enough. I would recommend showing one of the lines in black and the other in pink, as the authors have done.*

- **Dear Referee,**

**We agree on the Figures 5, 11 and B1 and we modified the lines more contrasted on revised version.**

*As for the results of the studies, I have a clear opinion that 40Ar/39Ar dating of fluid and mineral inclusions in quartz is absolutely unpromising from a geochronological point of view. In general, the authors write this in the discussion section, but in the final fourth conclusion they offer a geological interpretation that I do not think is supported by the data presented.*

- **Dear Referee,**

**Thank you for your comments regarding the implications section of our discussion. We acknowledge that the interpretation of our data, derived from $^{40}$Ar/$^{39}$Ar dating, is limited. However, we have endeavoured to provide the most plausible interpretations based on the available data.**

**We recognize the possibility of low-salinity Variscan remnant fluid circulation, as indicated by Kirnbauer et al. (2012). Furthermore, the high-temperature, low-salinity meteoric water circulation described by Schroyen & Muchez (2008) supports an interpretation involving convective cells of meteoric water circulation in study zone. In addition, we know that Virgo et al. (2013) showed that geomechanically veins can**

reactivate easily if their hosts are stronger than the veins themselves, which is the case in our study.

Nevertheless, despite the poor quality of the age results, they still suggest an overprinting event during the Mesozoic. This interpretation aligns with similarly ambiguous age signals reported from other areas of the North German Foreland Basin.

Taking your feedback into account, we have revised this section to present a more cautious interpretation of the data in the revised manuscript.

Thank you for your thoughtful review, which has helped us refine our discussion.

*The authors also write (lines 410-411):*

*"Unlike studies that obtained consistent ages from FIs (Qiu & Wijbrans, 2006; Qiu et al., 2011; Bai et al., 2013, 2019), we were unable to date FIs in Rursee quartz samples, likely due to high 40ArE concentrations and/or low salinity."*

*I, and probably other readers, would like to understand why other studies have succeeded in dating fluid inclusions and the authors have not. Is it a coincidence that such unsuccessful samples were found, or is the dating of fluid inclusions in quartz in principle impossible? A short sentence here is clearly not enough. I would like to see more generalisation. In other words, it would be useful to have a more detailed analysis of the work of predecessors where such dating was possible. Is it really possible, or did the authors of the earlier works draw too optimistic conclusions about such dating?*

- **Dear Referee,**

**Studies by Qiu & Wijbrans (2006), Qiu et al. (2011), and Bai et al. (2013, 2019) on fluid inclusions in wolframite and eclogite minerals have shown that these host minerals preserve primary fluid inclusions in growth zones as well as secondary fluid inclusions. Notably, these fluid inclusions exhibit high salinity (>20 eq. wt.% NaCl) and elevated potassium concentrations from KCl, which enable differentiation of fluid inclusion $^{39}$Ar spectra during crushing.**

**The key distinction between previous studies and our research, as discussed, lies in the recrystallization of quartz veins. This process results in the complete loss of the original high-salinity primary fluid inclusions in studied quartz samples. Consequently, we only observe low-salinity pseudosecondary and secondary fluid inclusions (3–8 eq. wt.% NaCl), which inherently contain lower K concentrations. Without sufficient K in the system, the $^{40}$Ar/$^{39}$Ar dating method cannot be effectively applied. Furthermore, given that recrystallization occurred at low temperatures (as inferred from the homogenization temperatures of pseudosecondary fluid inclusions), it is likely that this process reset the argon source, affecting any potential age determinations.**

In conclusion, while previous studies on fluid inclusions were well-conducted and provide realistic insights, the $^{40}$Ar/$^{39}$Ar dating method is not viable for low-salinity, low-potassium fluid inclusions, particularly when argon resetting occurs due to recrystallization.

**Authors response to Referee 2:**

- **Dear Referee,**
**Thank you for your contribution on our submitted manuscript, we appreciate deeply your comments and consider your valuable thoughts on our revised version of manuscript.**

*The crushing technique is very important to date fluid inclusions, and to exclude the trapped excess argon and obtain high-precision age messages for K minerals. The authors make a great effort to complete the research with more than sixty crushing steps (excluding the blanks) for each sample and relevant supportive analyses. The manuscript is well written and should be recommended for publication after modification. The main points include:*

1. *"To date, three main hypotheses are being debated as to the origin of the released argon in a stepwise crushing experiment. The first group (Qiu & Wijbrans, 2006, 2008; Bai et al., 2019) suggests that progressive crushing releases gases mainly from FIs and therefore represents FIs ages" (Lines 272 – 274). This group also suggested the radiogenic argon within the K-mineral lattices might be released by **prolonged crushing** when the grain sizes were reduced to tens of nanometers: Bai et al. (2018) proposed that "the K/Ca ratios of 09PT37Q in the final crushing steps are gradually rising up (Fig. 7c, purple lines), indicating gas release from a K-rich phase. Therefore, the gas mixing line is reasonable"; "The gas release sequence with sufficient crushing can be summarized as from microcracks → SFIs → PFIs → micrometer- to nanometer-sized minerals" (Bai et al., 2022); "The radiogenic argon ($^{40}Ar_R$) within the microcline lattices might be obviously liberated by prolonged crushing when grain sizes were reduced to tens of nanometers" (Bai et al., 2024).*

- **Dear Referee,**
**We appreciate deeply your comments and consider your valuable thoughts on our revised version of manuscript. Therefore, we add "The gas release sequence with sufficient crushing can be summarized as from microcracks → SFIs → PFIs → micrometer- to nanometer-sized minerals (Bai et al., 2022);" in our revised version.**

2. *"In the latter stages of the experiment (from the 20th analysing steps), the substantial release of $^{39}Ar_K$ isotopes may support the hypothesis proposed by Kendrick and Philips (2007) and Kendrick et al., (2011), suggesting the presence of K-bearing mineral inclusions in the samples and/or $^{40}Ar^*$ from the crystal lattice and also non-crushed small-sized FIs (<5 μm)" (Lines 372 – 374). I do not agree with this point. The K-bearing mineral inclusions in quartz should release the lattice argon in the final "pseudo-plateau" steps and a few earlier steps. The gas released in the first 20 steps should be from the secondary fluid inclusions. The reasons include: 1) Fig.3 shows that the quartz samples contain many fine fluid inclusions; 2) The sizes of K-bearing mineral inclusions should be tens of microns because the larger impurities had been excluded under binocular microscope; 3) A positive correlation between K and Cl is a characteristic of geofluids, not the solid minerals; 4) Your crusher with a pestle of 69.5 g hits sample more gently than that at CUG Wuhan (218*

*g) and much weaker than that of the air-actuated (90 psi) crushing device used by Kendrick and Phillips (2009).*

- **Dear Referee,**
**We respectfully disagree with your points regarding the degassing of K-bearing minerals only occurring during the final stage of the stepwise crushing process. Unfortunately, quartz grains that we analyzed are free from the primary fluid inclusion and they contain mostly pseudo- and secondary fluid inclusions. Knowing the low salinity of both FIs generation (3-8 eq. wt.% NaCl), K/Cl ratio should be mostly varied around ~1, knowing KCl from the salinity. However, we do not observe that after 20th analysing step. It is important to note that by the 20th pestle drop, there have already been over ~1000 impacts.**
**This differs from the findings of Bai et al. (2019), which resulted in different outcomes than expected from their previous work. Given this, we believe it is highly plausible that any occurring K-bearing minerals may degas together with the fluid phase.**
**We understand that about queries that our pestle weight is less heavier than Wuhan CUG pestle and much weaker than Kendrick and Philips (2009) crusher device to release radiogenic argon within the K-bearing minerals directly in the early stage of crushing, however considering that we have at the end of the stepwise crushing more than ~40 000 cumulative pestle drops which should not be dramatically differ from the above crusher system.**

3. *The K–Cl–Ar$^*$ correlation diagrams of ($^{40}Ar^*$/K vs Cl/K), (Cl/$^{40}Ar^*$ vs K/$^{40}Ar^*$) and ($^{40}Ar^*$/Cl vs K/Cl) are helpful to obtain the ages of secondary and primary fluid inclusions respectively, and to distinguish the gas sources.*

- **Dear Referee,**
**We appreciate your thoughtful comments. As per Bai et al. (2019), we generated all the relevant plots during our study; Unfortunately, none of the samples provided clear differentiation between the gas sources. We think that due to recrystallization, our vein quartz lost any primary high salinity brine. Given the resultant low concentration of K in the fluid, the chances of finding any correlation in the isotope correlation plots is low. We believe that, as mentioned in your second comment, there may be a mixing of fluid and solid sources after the 20th crushing step. An Excel file (and pdf as well) with all plots is attached with this file.**

4. *The total $^{39}Ar_K$ of R1 Rursee (Rursee 1b BNV?) in Supplementary 1.xlsx is only 89 fA, indicating too low K in the sample. This is the main reason that the samples do not yield good isochron ages.*

- **Dear Referee,**
**We agree that in our example we are limited, most probably due to the loss of the primary high salinity brine and consequent low K-contents of our fluid inclusions.**

**Minor points:**

1. *The unit of argon isotopes should be fA, not A in "Supplementary 1.xlsx" and "Supplementary 3.xlsx".*

- **Dear Referee,**
**We will correct that on our revised manuscript.**

2. *I do not understand the grain sizes in "Supplementary 4.xlsx".*

- **Dear Referee,**

**This is a grain size distribution analysis of the crushed quartz materials, aimed at determining whether all the fluid inclusions are decrypted during the crushing process or not. Unfortunately, after ~1000 (20th analyzing step) of pestle drop, most of the grain size were still higher than fluid inclusions sizes which revealed that fluid phase and solid phase after 20th analyzing steps released K simultaneously rather than fluid phase -> solid phase.**

*References:*
*Bai X.J., Li Y.L., Hu R.G., Liu X., Tang B., Gu X.P. & Qiu H.N., 2024, High-precision microcline $^{40}Ar/^{39}Ar$ dating by combined techniques: CHEMICAL GEOLOGY, 655. https://doi.org/10.1016/j.chemgeo.2024.122086.*
*Bai X.J., Liu M., Hu R.G., Fang Y., Liu X., Tang B. & Qiu H.N., 2022, Well-Constrained Mineralization Ages by Integrated $^{40}Ar/^{39}Ar$ and U-Pb Dating Techniques for the Xitian W-Sn Polymetallic Deposit, South China: Economic Geology, 117: 833–852. https://doi.org/10.5382/econgeo.4889.*
*Bai X.J., Jiang Y.D., Hu R.G., Gu X.P. & Qiu H.N., 2018, Revealing mineralization and subsequent hydrothermal events: Insights from $^{40}Ar/^{39}Ar$ isochron and novel gas mixing lines of hydrothermal quartzs by progressive crushing: Chemical Geology, 483: 332–341. https://doi.org/10.1016/j.chemgeo.2018.02.039.*
*Kendrick M.A. & Phillips D., 2009, New constraints on the release of noble gases during in vacuo crushing and application to scapolite Br-Cl-I and $^{40}Ar/^{39}Ar$ age determinations: Geochimica et Cosmochimica Acta, 73: 5673–5692. https://doi.org/10.1016/j.gca.2009.06.032.*

*Thank you very much for your rapid response to my comments. Obviously you did not discuss with your senior supervisors in such a short time.*

*The results of K-rich microcline by $^{40}Ar/^{39}Ar$ crushing indicated that a lot of gas was released from the microcracks and secondary fluid inclusions in the first several steps (Bai et al., 2024), implying that the gas trapped in fluid inclusions is much more than that released from the microcline lattices. The small mineral inclusions with your quartz samples contain much less potassium than microcline. You also admitted that "after ~1000 (20th analyzing step) of pestle drop, most of the grain sizes were still higher than fluid inclusions sizes", the gas from fluid inclusions would still dominate for more crushing steps until the very fine fluid inclusions (<1 μm) were exhausted.*

- **Dear Referee,**

**Thank you for taking time to evaluate our reply to your previous remarks. As we mentioned in our previous response,**

**1) the K/Cl ratio after the 20th step increases dramatically, reaching levels far higher than expected for fluid inclusions—at least 10 times higher suggesting that the chlorine containing brine reservoir has depleted by this time. This cannot be explained solely by the presence of low-salinity (3–8 eq. wt.% NaCl) fluid inclusions, as indicated by microthermometry data, and therefore we must postulate the contribution of another low-Cl K-bearing reservoir, i.e. other than the low salinity brine containing fluid inclusions**

**2) EPMA data show that K-bearing minerals (>10 μm) are significantly larger than fluid inclusions (~5 μm). By step 20 of the crushing experiment the average particle size has been reduced to <10 μm, and therefore we have started to reduce the grain size of K-bearing minerals inclusions (e.g. mica) in this phase of the experiment. Consequently, solid phase reservoirs have the potential to degas simultaneously after 20th with fluid phase reservoirs.**

**3) Calculations from the Supplementary file indicate that K-bearing minerals and the crystal lattice of quartz together may retain a substantial proportion of the K-concentration during stepwise crushing.**

**These three main arguments suggest that in the absence of high salinity brine containing fluid inclusions, K-bearing solid phase reservoirs may begin to degas from the 20th step onward, while fluid-phase reservoirs that contain low salinity fluids decrease. As stepwise crushing progresses, the solid-phase contribution is likely to increase, and thus after the the reservoirs containing fluid phases have largely been exhausted, K is contributing to the observed signal, and hence is likely to have come from solid phases.**

**Our study may differ from that of Bai et al. 2024's in that in our case we dealt with fluid inclusions of low salinity, whereas Bai et al. 2024 although this wasn't explicitly mentioned may have worked with samples with high salinity brines. If this was indeed the case, their observations may have been fundamentally different from ours.**

**Authors response to flash review of Associate Editor:**

*"Dear Dr Huseynov,*

*I have now received recommendations from two reviewers. Both reviewers agree that the data and the main questions you attempt to address is interesting and they are both supportive of publication in GChron pending some revisions. You will also find my own flash review below which require minor revisions as well, not so much on the text / interpretation, but on some definitions you use.*

*The two reviewers and myself altogether made a list of comments that should help you improve some aspects of your manuscripts.*

*Therefore I am recommending the equivalent of moderate revisions and request that you first provide responses to the reviewers and myself on how you will address those points (aka a rebuttal letter). Should you disagree with any of the reviewer's comments you would need to justify your choices in your responses. I saw that you responded to most comments from the two reviewers already. If you want to add some additional points, please make it so by responding to this comment along with your response to my comments.*

*Note that the Peer-review process in GChron is different than standard journals in that, only after I have read your final responses / rebuttal letter, I will either invite you to submit a revised manuscript or directly reject the manuscript in the case that your responses are not satisfying. That being said, nothing prevents you to work on the manuscript as you elaborate your answers in the anticipation of a positive outcome.*

*Fred"*
* * *
**Dear Editor,**

**We would like to thank you for your time and consideration. We have carefully reviewed both referee comments and we have addressed them accordingly. While we agreed with most of the suggestions and we have incorporated them into our revision, we respectfully disagree with some of the others, as outlined in our response. We added also our response to your comments below.**

**We would like to emphasize that prof. Ivanov and Qiu each represent two fundamentally different and opposing schools of thought. Prof. Ivanov has expressed scepticism regarding the effectiveness of dating by crushing, whereas prof. Qiu has built his career on this method. In our study we critically evaluate the effectiveness of the approach in our case. It follows logically that we agree and disagree with points raised by both reviewers. We have decided to follow prof. Qiu's advice, given his extensive experience in this field. However, we note the limitations in our case, particularly due to the suspected low concentrations of potassium. As prof. Qiu also noted, this is likely a consequence of the loss of primary high-salinity brines through recrystallization.**

We have carefully considered the comments of prof. Qiu about different step of gas release, which was the main discussion with prof. Qiu in our earlier revision. We think that low peaks (~1080 cm$^{-1}$) shown by Raman may correspond to other components of K (as $K_2CO_3$) in fluid inclusions and these could change the K/Cl ratio dramatically. Therefore, the release of $^{39}Ar_K$ maybe associated with first stages of release derived from a secondary fluid inclusion reservoir, subsequent release from the 20th step onward by pseudo-secondary fluid inclusions and at the end of experiment gas is most probably released from the crystal lattice or K-bearing mineral inclusions as prof Qui suggested in the open discussion. So, we will add this to our discussion part as a first option and also, we will keep our original interpretation as a second option.

We appreciate your consideration of our revised manuscript and look forward to your feedback.

Akbar A.O. Huseynov,

on behalf of the co-authors

*Flash review by AE – F. Jourdan*

*This topic fits well in this journal. I like it.*

*L14 (and rest of abstract). Please, write fluid inclusion in full for the abstract (and possibly for the rest of the text). It's not a standard abbreviation like MORB for example and texts are always easier to read without abbreviation that could be avoided.*

- Dear Editor,

We will modify our revised version accordingly.

*L43. That makes sense that meteoric water would have an atmospheric ratio, but note that also hydrothermal waters can have sub-atmospheric ratios as well and that can be seen sometimes in the inverse isochron of fluid altered rocks (e.g. 290 – 280; Cf. Baksi, 2006) which I observed so many time for altered basalt or plagioclase. May or may not be relevant here, but I though I would just mention that for completeness.*

- Dear Editor,

We will add your suggestion in our revised version accordingly.

*L49 – Rama, 1965. Well, I don't think the method was available back then, more like a 70's technique, so I question this reference, or a typo in the year.*

- Dear Editor,

**Indeed, Rama et al. (1965) used K-Ar and not $^{40}$Ar/$^{39}$Ar. We therefore propose the following sentence for the revised manuscript:**

**It has been posited that the presence of $^{40}$Ar$_E$ in fluid inclusions could create a challenge to determining accurate vein formation ages using the K-Ar dating technique (Rama et al., 1965). More recent, isochron diagrams using $^{40}$Ar/$^{39}$Ar geochronology help to overcome this issue (McKee et al., 1993; Qiu, 1996; Qiu et al., 2002).**

*Fig. 4 and throughout the text – I have a big problem with you defining ages based on totally discordant steps. I totally understand that the crushing technique differs from the standard laser / furnace approaches and only lead useful results toward the end of the experiment (hence a much shorter step-concordant section than what is usually expected), but having discordant steps and calculating an average based on that, could not be further for obtaining an age. An age provides an information on an event that happen precisely at the time it gives within error. In the first spectrum (1a BNV) the steps are still decreasing and are not concordant with each other. This can be quickly verified by the large MSWD values from your table which attest to discordant steps. So do you think there is any meaning to this number within uncertainties? I think not at all. Although some other convergent portions are better than other, they all show a decreasing pattern. What that tells me is that you have maximum (error / apparent) ages. So I would like you to refrain using the word age by itself implying a event that occurs precisely at this time. you can use maximum apparent age, or maximum error age, but not "age" alone as this is incorrect.*

**- Dear Editor,**

**The MSWD values indeed indicate that some plateaus by stepwise crushing are better than others. We only consider maximum apparent ages from Rursee 1b BNV, Rursee 2 BPV and isochron apparent age of Rursee 4 BPV as geologically meaningful as they did not reveal any pseudo-plateau by stepwise crushing during the experiment like the other samples. We agree that use of age alone in this context is somewhat confusing, therefore we will use maximum apparent age throughout revised manuscript.**

*Fig. 4 and in text – The error age are given with too many digits. It looks like it comes from the spreadsheet without any though on what they mean. Considering that they are error ages and the relatively low precision, they should be written like the format 144 ± 7 Ma for most of them, and possibly 96.6 ± 2.1 for the most precise (although unnecessary since max age)*

**- Dear Editor,**

**We will modify the number of digits in our revised version accordingly including figures and tables.**

*Fig 5. – please indicate the intercept ratios and maybe the error ages associated with them.*

**- Dear Editor,**

**We will add these ratios and error ages to our revised version.**

*L272 – There is rigorous definitions of what should be call a plateau in the literature, there is no plateau whatsoever in any of your dataset, so maybe call it something like converging section, or something like that.*

**- Dear Editor,**

**We agree with your comments, therefore we accept to use "late converging section" which we define as a plateau like segments with a reduced spread in ages.**

*L398 – calling it a "pseudo plateau" is a noble effort, but again, the steps don't even overlap, so there is no flat (hence plateau) section, rather the steps converge, so a converging section would be more of an appropriate term. Especially important since you really tell yourself in the discussion / conclusion, that you did not obtain any meaningful ages.*

**- Dear Editor,**

**We agree with your comments. We propose to indicate the "pseudo plateau" now for our study as an "early converging section", whereas the "plateau" will be referred to as a "late converging section" to differentiate between the true plateau and the more disturbed results.**